# Ocean acidification conditions increase resilience of marine diatoms

Jacob J. Valenzuela[1], Adrián López García de Lomana [1], Allison Lee[1], E.V. Armbrust[2],
Mónica V. Orellana[1,3] & Nitin S. Baliga[1,4,5,6]

The fate of diatoms in future acidified oceans could have dramatic implications on marine ecosystems, because they account for ~40% of marine primary production. Here, we quantify resilience of *Thalassiosira pseudonana* in mid-20th century (300 ppm $CO_2$) and future (1000 ppm $CO_2$) conditions that cause ocean acidification, using a stress test that probes its ability to recover from incrementally higher amount of low-dose ultraviolet A (UVA) and B (UVB) radiation and re-initiate growth in day–night cycles, limited by nitrogen. While all cultures eventually collapse, those growing at 300 ppm $CO_2$ succumb sooner. The underlying mechanism for collapse appears to be a system failure resulting from "loss of relational resilience," that is, inability to adopt physiological states matched to N-availability and phase of the diurnal cycle. Importantly, under elevated $CO_2$ conditions diatoms sustain relational resilience over a longer timeframe, demonstrating increased resilience to future acidified ocean conditions. This stress test framework can be extended to evaluate and predict how various climate change associated stressors may impact microbial community resilience.

[1] Institute for Systems Biology, Seattle, WA 98109, USA. [2] School of Oceanography, University of Washington, Seattle, WA 98105, USA. [3] Applied Physics Laboratory, Polar Science Center, University of Washington, Seattle, WA 98105, USA. [4] Departments of Biology and Microbiology, University of Washington, Seattle, WA 98195, USA. [5] Molecular and Cellular Biology Program, University of Washington, Seattle, WA 98195, USA. [6] Lawrence Berkeley National Lab, Berkeley, CA 94720, USA. Correspondence and requests for materials should be addressed to M.V.O. (email: morellana@systemsbiology.org) or to N.S.B. (email: nbaliga@systemsbiology.org)

By the end of the twenty-first century atmospheric $CO_2$ is expected to reach 800–1000 ppm[1], with a corresponding drop in ocean pH of 0.3–0.4 units, a phenomenon termed "ocean acidification." Concomitant with a rise in temperature, shoaling of the mixed layer, and a decline in nutrient availability, ocean acidification is predicted to have dramatic impacts on marine food webs[2,3]. For instance, coccolithophores are predicted to become more sensitive to ocean acidification[4], because it will synergistically exacerbate their susceptibility to other stressors such as ultraviolet radiation (UVR)[5]. However, the intricacies of elevated $CO_2$ on marine carbon chemistry and its effect on coccolithophores have yielded varied results[4,6] and highlights the complexity of ocean dynamics. Ocean acidification is also predicted to cause a decline in the bioavailability of iron (Fe), thereby imposing additional stress on phytoplankton populations[7].

As dominant primary producers in marine ecosystems[8], diatoms are a vital base for the food chain and the biological carbon pump—a complex set of interactions among diverse microorganisms that transports fixed carbon from surface phototrophic zones to the deep ocean[9]. Although species composition and abundance of diatoms are known to fluctuate with environmental changes, their fate in acidified oceans of the future is uncertain[2,10–13]. Thus, diatoms are an integral group for monitoring environmental conditions. They are known to have been impacted by prior climate shifts and have been used as record keepers of Earth's past conditions[14]. For example, species composition and isotopic signatures of diatom microfossils have been used to infer nutrient conditions during the last ice age[15] and trace how biological carbon export production changed throughout the glacial and interglacial cycles[16]. Environmental histories can be inferred using simple transfer functions that correlate diatom sensitivity to specific factors such as $CO_2$ levels[17], sea surface temperatures[18], upwellings, eutrophication events[19,20], and even dynamics of ocean currents[21]. However, it is complicated to predict how diatoms will respond to future climates by simply correlating the consequence of changing one or a few environmental conditions on growth. Moreover, improved growth rate[22] or increased biomass accumulation under higher $CO_2$ conditions does not necessarily mean in itself that diatoms will be less sensitive to ocean acidification, that is, more stable.

The concept of "ecological resilience" may be a more apt metric of stability and is defined as "the amount of disturbance that can be tolerated by a system without changing state and still persist"[23–26]. The more resilient a system, the larger is the disturbance needed to force it into an alternate stable state[24,27]. We developed a stress test to quantify resilience of diatoms under simulated mid-twentieth century and future oceanic conditions, by assaying their ability to tolerate and recover from progressively larger amounts of stress. The primary objective of the stress test was to investigate whether growth under elevated $CO_2$ would alter diatom resilience. For instance, elevated $CO_2$ reduces the need for biophysical carbon concentrating mechanisms (CCMs) to saturate RuBisCO (ribulose-1, 5-bisphosphate carboxylase oxygenase) at approximately 80%, while minimizing diffusion of $CO_2$ back across membranes[28–34]. When ambient $CO_2$ doubles, it is estimated that the resulting downregulation of CCMs conserves 3–6% of the energy required for carbon fixation in diatoms[28]. We hypothesize that the predicted energy savings during growth under elevated $CO_2$ could be reallocated for state transitions or potential stress management and thereby improve the resilience of diatoms in future oceanic conditions.

The diatom Thalassiosira pseudonana has evolved in a dynamic environment experiencing daily and seasonal changes with both predictable and unpredictable fluctuations in diverse environmental factors including light, pH, temperature, salinity, and micronutrient availability. Accordingly, T. pseudonana can adopt at least four principal physiologic states matched to four key environmental conditions (light, dark, nutrient replete, and nutrient deplete), and it possesses regulatory networks to mediate transitions between these states[35]. Energetically expensive mechanisms (transcription, translation, stress response, etc.) are required to both sustain an environmentally relevant physiological state in face of small perturbations and to drive transition to a new state in response to, or in anticipation of[36], an environmental change[35]. When faced with a fluctuating environment, the success of an organism depends on its ability to efficiently manage trade-offs between allocating resources for producing biomass and transitioning between alternate physiological states, while mitigating consequences of the associated stress[37,38]. We speculated that within such a fluctuating environmental framework, any stabilizing or destabilizing effect of ocean acidification on the diatom T. pseudonana should be detectable in terms of altered dynamics of its capacity to re-establish growth from incrementally increasing stress[39], and its subsequent ability to restore and maintain routine transitions across the aforementioned four physiologic states.

We investigated the stability and the response and recovery dynamics of T. pseudonana in mid-twentieth century $CO_2$ conditions (300 ppm $CO_2$; heretofore low carbon or "LC" condition) and projected end-of-twenty-first century conditions (1000 ppm $CO_2$; heretofore high carbon or "HC" condition), while transitioning between the routine fluctuations of four physiological states (light, dark, nitrogen replete and nitrogen deplete). Culturing in nitrogen-limiting conditions imposes an added requirement that diatoms shift from assimilating externally available nitrate in early phases of growth to recycling internal nitrogen reserves at later stages when they reach carrying capacity[35,40]. In addition to deceleration in growth rate, this shift is accompanied with a reduction in photosynthetic efficiency[40,41], thereby requiring coordination across diurnally and nutritionally modulated physiological states[35]. Thus, we speculated that the energetically expensive acclimation to continually fluctuating environments might manifest in differences in response and recovery dynamics between the LC and HC cultures.

## Results

**Dynamics of non-UVR growth transitions.** Thalassiosira pseudonana was batch cultured in triplicate photo-bioreactors with nitrogen-limiting conditions (~65 μM nitrate), 12:12 h light:dark (L:D) cycles, and saturating photon flux density (275–300 μmol photons $m^{-2} s^{-1}$). At the end of each growth cycle or "stage" (i.e., when the culture reached late exponential phase and nitrogen level was below detection limit (~1 μM nitrate)), a small culture aliquot was transferred into fresh nutrient replete medium to re-initiate growth (Fig. 1). Serial transfers were performed over four stages of culturing in the two $CO_2$ conditions to assess whether cells within an aliquot from the preceding stage had the capability to recover from stress, re-initiate population growth, and transition to an environmentally appropriate physiological state. Both LC and HC cultures maintained consistent growth rates through all stages, with higher carrying capacities under HC conditions (Fig. 1a, b). Relative to HC cultures, growth dynamics of LC cultures became more variable across replicates in latter stages with an increased lag and a lower carrying capacity in stage 3 (Supplementary Fig. 1). This is interesting because it has been shown that dilution of a resource-limited population amplifies variability across replicate cultures, and when pushed to the extreme with progressively greater stress in each stage, such cycles of continued dilution may drive replicate cultures to collapse at different times depending on their respective histories[39,42–45]. The increased variability in growth dynamics of LC cultures

suggested that *T. pseudonana* is more sensitive to fluctuating environments in LC conditions and, therefore, potentially more susceptible to the addition of another ecologically relevant stress.

**Response and recovery dynamics during UVR stress test.** We therefore subjected LC and HC grown *T. pseudonana* cultures to progressively higher doses of UVR insults as a stress test designed to amplify differences in response and recovery dynamics (i.e., resilience). Exposure to UVR in the photic zone can damage photosystem II (PSII) and inhibit photosynthesis in phytoplankton[46–49]. UVR also inflicts molecular and cellular damage[50] directly through the formation of cyclobutane pyrimidine dimers in DNA[47] and indirectly through generation of reactive oxygen species (ROS)[48]. The combined effects of elevated $CO_2$ and UVR are complex and directional, whereas acclimation to UVR reduces susceptibility to photoinhibition of diatoms at high and low $CO_2$ conditions; acclimation to elevated $CO_2$ has been shown to

increase their sensitivity to photoinhibitory UVR[46]. In the first stage of the stress test, diatom cultures did not receive any UVR, but in subsequent stages, each culture was subjected to an incrementally higher dose of UVR ($0.5 \, \mathrm{mW \, cm^{-2}}$) for 1 h in the middle of the light cycle (Fig. 1c, d).

Under LC conditions, *T. pseudonana* survived 9 L:D cycles over two stages and five 1 h exposures of $0.5 \, \mathrm{mW \, cm^{-2}}$ UVR. None of the LC cultures recovered from the second dilution event and all failed to re-initiate population growth upon transitioning into the third stage in which the UVR dosage was increased to $1.0 \, \mathrm{mW \, cm^{-2}}$ (Fig. 1c). In contrast, two HC cultures recovered from the second stage after a 5-day lag and grew for an additional 4 L:D cycles, withstanding eight additional exposures to a higher dose of UVR in the third stage ($1.0 \, \mathrm{mW \, cm^{-2}}$) (Fig. 1d). Under both LC and HC conditions, cultures took longer to recover in each subsequent stage until they could no longer resume growth (lag times increased from ~1 to 2 days in the 1st and 2nd stages of LC

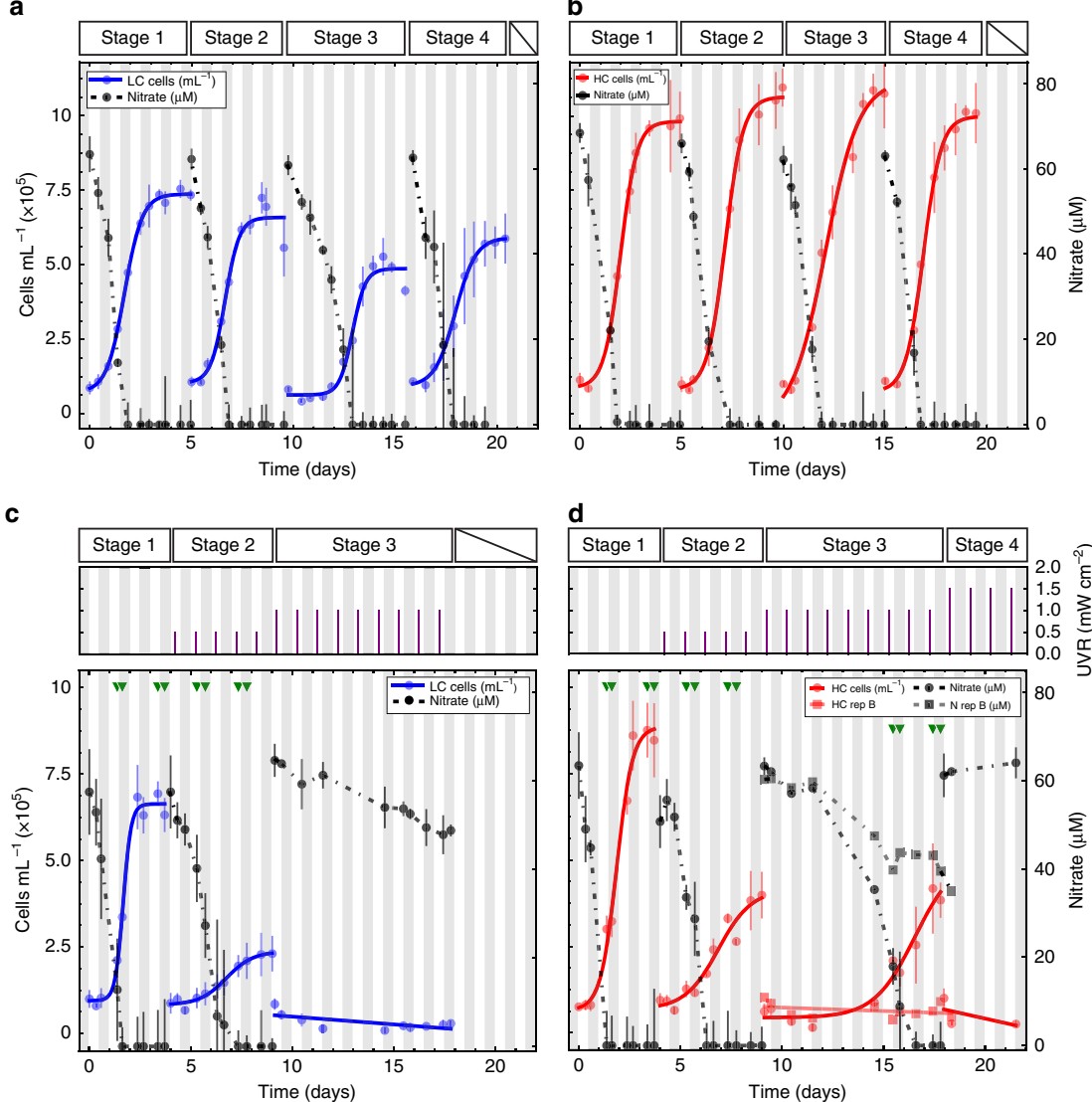

**Fig. 1** Growth dynamics of *T. pseudonana* at LC and HC conditions during non-UVR growth transitions and the stress test. Growth characteristics of *T. pseudonana* cultures without UVR under LC (300 ppm $CO_2$; **a**) and HC conditions (1000 ppm $CO_2$; **b**). Number of cultures across all non-UVR stages ($n_{LC \, \text{stage} \, 1,2,3,4} = 3$; $n_{HC \, \text{stage} \, 1,2,3,4} = 3$). During the stress test cultures at LC (**c**) and HC (**d**) conditions received progressively higher doses of UVR in each stage ($n_{LC \, \text{stage} \, 1,2,3} = 3$; $n_{HC \, \text{stage} \, 1,2} = 3$; $n_{HC \, \text{stage} \, 3,4} = 2$); see Methods for detailed experimental design. Dashed black lines represent the decrease in nitrate levels to below detection (<1 μM). Vertical error bars denote the standard deviation of the mean. White and gray bars in the background indicate the 12 h light and 12 h dark phases of the diurnal cycle, respectively. Purple lines indicate UVR dose in mW $cm^{-2}$. Green triangles mark time-points at which transcriptomic analyses were performed

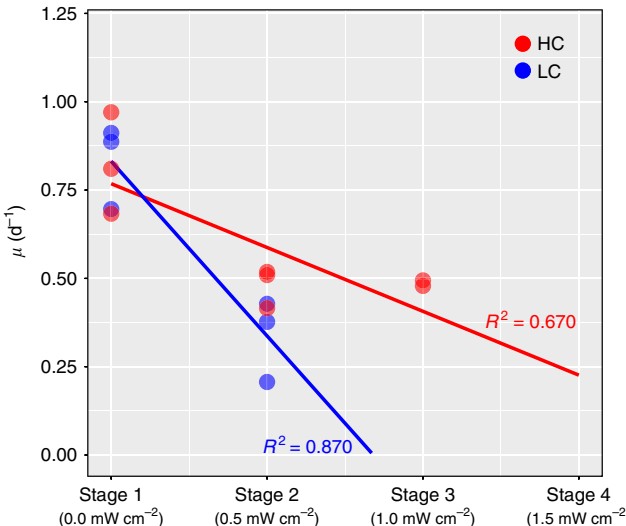

**Fig. 2** Effect of $CO_2$ levels on growth rate dynamics of LC and HC cultures in stress test. Specific growth rate of diatom cultures growing in LC conditions declines faster when subjected to incrementally higher UVR stress, relative to diatom cultures growing in HC conditions. Circles depict specific growth rates for replicate cultures in HC (red) and LC (blue) conditions in stages 1 and 2 ($n = 3$), and HC cultures in stage 3 ($n = 2$). Lines represent linear regression of the HC and LC data points

conditions, respectively, and ~0.5, 1.5, and 5.5 days in the 1st, 2nd, and 3rd stages of HC conditions, respectively). As a control, the inoculum from each transfer was split into two sets of reactors. One set received UVR, whereas the other did not (Supplementary Fig. 2), demonstrating that UVR alone was the cause of impaired growth, and eventual collapse. Importantly, the range of UVR doses across all stages were not lethal (Supplementary Fig. 3) and comparable to UVR levels experienced by diatoms and other phytoplankton in the natural environment[1,5,49,51]. Exposure to a daily dose of 0.5 mW cm$^{-2}$ of UVR for 1 h at mid-day decreased the specific growth rate to a greater extent in LC than in HC conditions (Fig. 2). Combined, these results demonstrate that the diatoms were more resilient[39,42,43] under HC conditions with increased capability to recover from repeated exposures to higher doses of UVR. The energy trade-off required to sustain routine transitions across light/dark and nitrogen replete/deplete physiological states increased the susceptibility of LC grown *T. pseudonana* to UVR, and revealed that growth under HC conditions contributes to higher resilience of diatoms.

**Differential gene expression analyses**. We performed whole transcriptome RNA-seq analysis of replicate cultures subjected to the stress test under the two $CO_2$ conditions to investigate if differential gene expression would yield a mechanistic explanation for why low doses of UVR in conjunction with periodic physiological state transitions drives diatom cultures under LC conditions towards collapse sooner than cultures growing in HC conditions. The transcriptome sampling schedule captured transitions of the diatom cultures across the four physiologic states. We performed targeted analysis of the transcript level changes in genes that are putatively associated with UVR-generated stress and carbon metabolism. We analyzed transcript levels of 38 putative oxidative stress response (OSR) genes (i.e., peroxidases, glutaredoxins, and superoxide dismutases) that are known to mitigate cellular damage caused by ROS generated by UVR exposure (Supplementary Fig. 4). Because UVR also damages DNA directly through the formation of cyclobutane pyrimidine

dimers[47,50], we also analyzed transcript levels of 69 putative DNA repair genes, including those involved in homologous recombination, base excision repair, mismatch repair, nucleotide excision repair, and DNA photolyases (Supplementary Fig. 5). Across all of these analyses, we did not observe meaningful changes in transcript levels of OSR, DNA repair, central carbon metabolism, and light-harvesting complex genes. In fact, there was no discernible difference in the distribution of relative expression changes of UVR response and central carbon metabolism genes in HC vs. LC conditions between samples that received UVR and those that did not (Supplementary Figs. 6–8).

Recent studies performed by Clement et al.[33,52] under an acute $CO_2$ shift from 20,000 to 50 ppm, under constant light, discovered that CCMs are subject to strict regulation as a function of $CO_2$ concentration[33,52]. At low $CO_2$ conditions *T. pseudonana* utilizes a biophysical CCM to actively uptake $CO_2$ and bicarbonate, whereas at high $CO_2$ concentrations passive diffusion of $CO_2$ across membranes is sufficient to support photosynthesis. We analyzed and compared expression changes of 36 transcripts across HC and LC conditions to the proteomic response observed by Clement et al.[33] and identified a small set of genes that were downregulated under HC conditions (Supplementary Fig. 9). Two of these genes are associated with carbon acquisition and the other encodes the "low $CO_2$ inducible protein of 63 kDa" (LCIP63: 264181), which has been reported to be expressed only when $CO_2$ is limited, and speculated to play a wider role in how diatoms respond to $CO_2$[33] (Supplementary Fig. 10). In addition, 13 of 21 genes within a $CO_2$-responsive gene cluster discovered previously by Hennon et al.[34] (CCM/PR 332) were significantly downregulated under HC conditions ($p$ value <0.001). The downregulated genes included bestrophin-like proteins, which share homology with a family of anion-selective channels[53] permeable to bicarbonate[54], as well as carbonic anhydrases[34] (Supplementary Table 1). Thus, the differential expression analysis did not implicate any particular gene or process in causing collapse, but it did suggest that downregulation of CCMs and related processes could have contributed to increased resilience under HC conditions.

**Physiological signatures of population collapse**. Since collapse and resilience are both emergent systems-level phenomena, we searched for predictive signatures of collapse at a global transcriptomic level. We observed high similarity of light/dark and early/late growth phase-specific transcriptomes across replicate cultures throughout stage 1 in both LC and HC conditions (Fig. 3). Variability across replicate transcriptomes increased in the stage prior to collapse, that is, in stage 2 for LC conditions and stage 3 for HC conditions (Fig. 3c). To understand the implication of this increased variability from a physiological perspective, we analyzed transcriptomes from the first stage (no UVR) across both $CO_2$ levels to identify genes associated with physiological state-characteristic transcriptional signatures. Altogether, 58 and 218 genes had distinct non-overlapping differential expression patterns during the light/dark and early/late phases of growth, respectively (see Methods, Supplementary Fig. 11, and Supplementary Data 1). These genes or "state descriptors" were used to generate a two-dimensional (2D) physiological state space in relation to the diurnal cycle (light or dark) and phases of growth (early or late). The physiological state of cells within each culture at a given time-point was elucidated by mapping their transcriptome into this 2D state space based on expression patterns of state descriptors (Fig. 4a, b). Using the state descriptors, we also mapped the growth states of *T. pseudonana* from a previous study which tracked transcriptome changes through all phases of growth at 400 and 800 ppm $CO_2$[35]. The growth conditions,

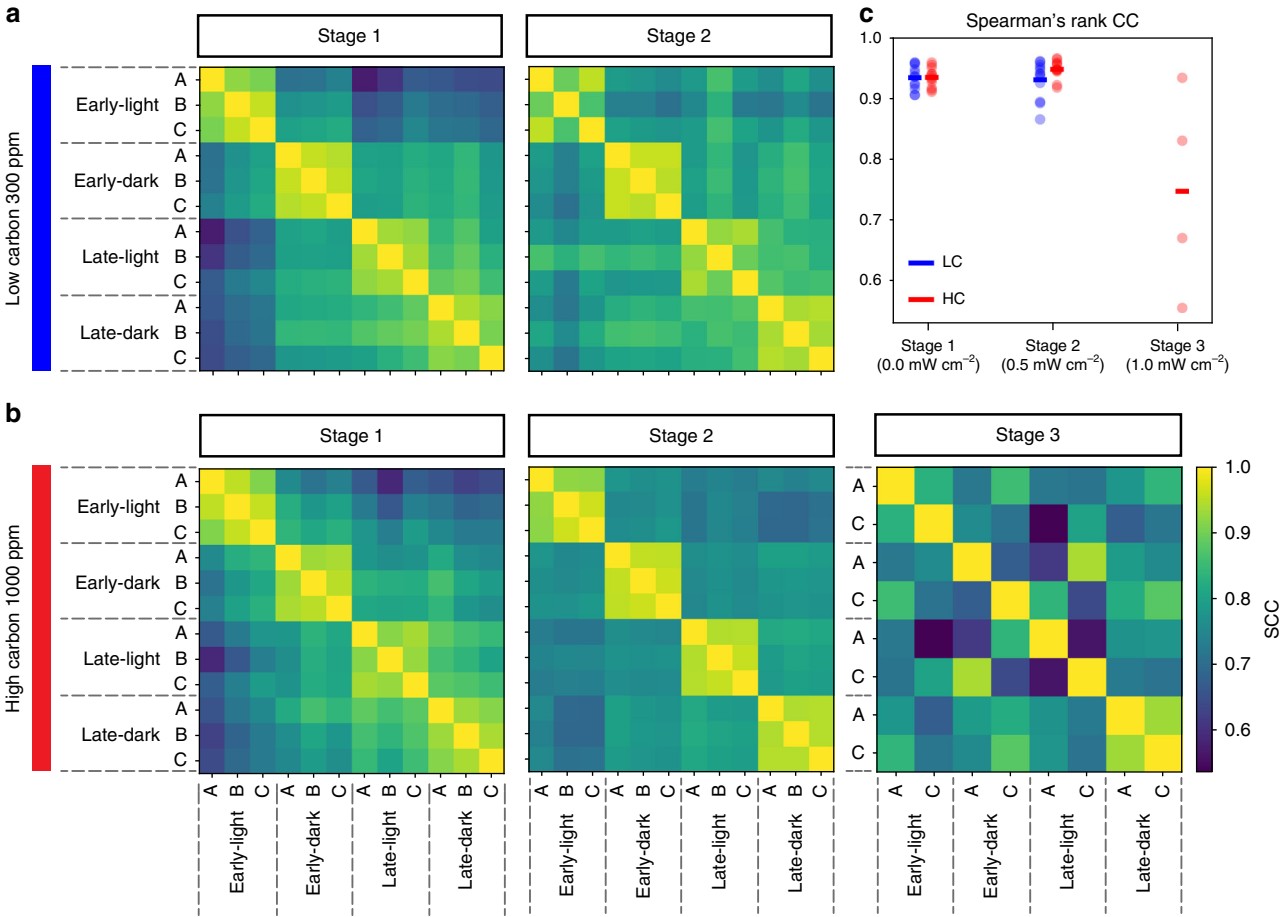

**Fig. 3** Replicate variability in diurnal cycle and growth phase transcriptome states prior to collapse. Spearman's rank correlation coefficients (SCCs) between transcriptomes across different L:D cycles, phases of growth, and stage of experiment in LC (**a**) and HC (**b**) conditions exposed to incrementally higher UVR doses during the different stages. SCC heatmap shows high similarity of transcriptomes from similar environmental conditions across stage 1, while similarity decreases in stages 2 and 3 of LC and HC conditions, respectively (see color bar for scale). **c** Dotplot of SCCs shows that transcriptomes became less correlated to each other in the stage prior to collapse (i.e., stage 2 for LC conditions and stage 3 for HC conditions)

sampling schedules, and transcriptome analysis platforms used in those experiments were different from our current study (Supplementary Table 2). However, despite these differences, we observed that the state descriptors accurately recapitulated how *T. pseudonana* reproducibly transitioned between light and dark states, while gradually moving through different growth phase-specific sub-states due to depletion of multiple nutrients (Supplementary Fig. 12).

Transcriptomes from the first stage (no UVR) of the stress test (irrespective of $CO_2$ levels) mapped to the four corners of this 2D state space, representing the four physiological states: light (early and late) and dark (early and late). In contrast, a small dose of UVR in stage 2 disrupted the expected genome-wide transcriptome changes, demonstrating that the diatoms were less efficient in transitioning through and adopting appropriate physiologic states. This decreased capability to restore the physiological state to the corresponding environment appeared much earlier (in stage 2) under LC conditions, relative to HC cultures (Fig. 4c, d). As the cultures approached collapse, the transitions between physiological states of replicate cultures became significantly more variable (Supplementary Fig. 13 and Table 3) and uncorrelated with the environment (i.e., they lost relational resilience)[27]. When microbial communities reproducibly adopt a function that is matched to conditions experienced in their environment, they are said to exhibit relational resilience[27] (e.g., when diatoms shift to a physiologic state appropriate for

supporting activities related to photosynthesis upon experiencing (or anticipating) a transition from nighttime to daytime)[35]. Upon losing this property, complex systems become sensitive to critical transitions[27] and may exhibit slow or irreversible recovery from stress. Loss of relational resilience occurred in the stage just prior to collapse—in the second stage for LC cultures and in the third stage for HC cultures. This loss of relational resilience at the transcriptome level manifests in increased variation in photosynthetic efficiency of the cultures through transitions across the four physiological states within each stage. While there were minimal changes in the coefficient of variation (CV) of $F_v/F_m$ (see Methods) across all replicates from cultures with no UVR stress (Fig. 5a), the CV of UVR-stressed cultures increased significantly in the stage preceding collapse (Wilcoxon signed-rank test; $p$ value <0.005), occurring in stage 2 under LC and stage 3 under HC conditions (Fig. 5b), further demonstrating that diatoms are less susceptible to additional stress in HC conditions relative to LC conditions. Thus, an increased CV in photosynthetic efficiency over time is potentially an early warning sign for an unstable diatom population. This increased variability is a characteristic property of complex systems on the brink of collapse[39,42,55], and it suggests that collapse of diatom cultures might not be a consequence of the malfunction of a specific gene or process, rather it is an outcome of system failure that emerges from global dysregulation of cellular processes. In other words, our data show that diatom cultures are prone to collapse when

their internal transcriptome state, and by proxy their physiological state, is uncorrelated with the external environmental condition.

## Discussion

The stress test developed in this study can quantify resilience of diatoms at a phenotypic level by monitoring trends in dynamics of response and recovery to UVR, variability in photosynthetic efficiency, and concordance of internal transcriptome state with external environmental conditions. The test builds on the observation that although the diatom cultures were able to sustain repetitive transitions among four physiologic states (Fig. 1a, b), just a low sub-inhibitory dose of UVR disrupted relational resilience and caused system failure (Fig. 1c, d, Fig. 4, and Supplementary Fig. 13). The diatoms were taxed by having to differentially regulate thousands of genes in order to continually transition among the four diurnally and nutritionally modulated physiologic states[35]. Further, we also present evidence that

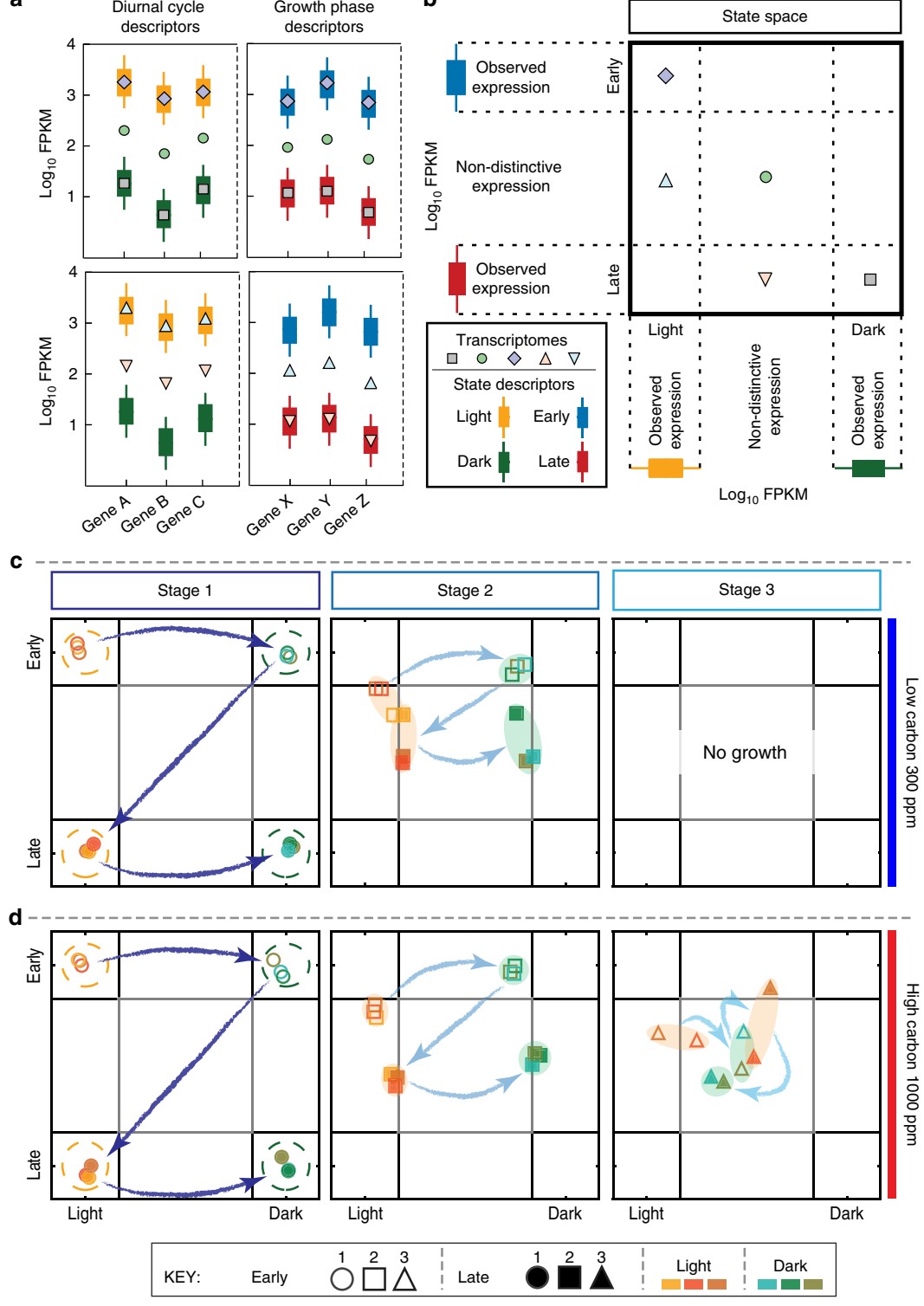

relative to LC conditions, diatoms growing under HC conditions were able to sustain over a significantly longer timeframe their ability to recover from UVR and adopt a physiologic state that was matched to the external environment.

We investigated global physiological consequences by analyzing genome-wide transcriptome changes, in an attempt to discover plausible mechanisms that could have contributed to resilience and collapse. The collapse phenomenon could not be attributed to UVR-induced altered expression of any particular gene or pathway associated with stress response (OSR and DNA repair), light-harvesting, and central carbon metabolism. However, there was evidence that the improved resilience of diatoms under HC conditions could have resulted in part from the reduced requirement of CCMs[28,34] and other carbon assimilation pathways[22,32,33,52,56,57], as well as the global response to growth at elevated $CO_2$. Saturating $CO_2$ levels to approximately 80% to enable carbon fixation by RuBisCO is energy intensive as protons and inorganic carbon have to be transported across the membrane against a gradient, while preventing diffusion of $CO_2$[28,29]. This includes the energy investment into producing the catalytic machinery as well as its operating costs[30]. Downregulation of CCMs in response to higher $CO_2$ concentrations has been estimated to save 3–6% of energy expenditures for carbon fixation[28]. Wu et al.[22] in 2014 performed an elegant experiment to test the consequence of such energy savings on five large diatom species. The authors observed that at increased $CO_2$ concentrations, PSII electron transport rates were unaffected, nor was their elemental stoichiometry, but they did observe elevated growth rates. Moreover, Wu et al.[22] did not observe any difference in light capture dynamics and photosynthetically produced reductants under varying $CO_2$ concentrations, which led them to conclude that the enhanced growth may be a result of increased diffusion rates of $CO_2$ and a lower metabolic cost due to downregulation of active carbon acquisition under elevated $CO_2$ conditions. These results support the hypothesis that reduced need for energy intensive processes such as CCMs under elevated $CO_2$ may contribute to increased resilience of *T. pseudonana* in future ocean conditions.

Consistent with our findings, abundance of a *Thalassiosira* spp. increased significantly from 14% to 37% of the phytoplankton composition of a natural community, when they were subjected to elevated $CO_2$ and UVR in a mesocosm study[58]. Increased $CO_2$ availability, however, is not the only change that diatoms will experience in future oceans. Decreased Fe bioavailability in an acidified ocean could have an opposite effect by increasing stress and inhibiting growth of *T. pseudonana*[7]. Given the variability in sensitivities of different phytoplankton and even species to different factors such as UVR, temperature, or nutrient limitations[58,59], a multi-factorial experiment is necessary to understand the complex trade-offs in dealing with these combinatorial environmental changes. The framework and tools developed in this study can be adapted for conducting such stress tests with more factors and on a broader range of organisms in order to assay and predict how natural marine microbial populations will fare in response to ocean acidification.

## Methods

**Batch culture growth, monitoring, sampling, and analysis**. All experiments were performed with the model diatom *T. pseudonana* (CCMP1335) grown in custom-made 1.5 L photo-bioreactors with enriched artificial seawater medium[60] modified to have reduced levels of nitrate (~65 μM). Axenic *T. pseudonana* cultures were acclimated to 300 and 1000 ppm $CO_2$ (mixed air) under a 12:12 h L:D regime for two consecutive growth cycles before being transferred to sterile nitrate-limited photo-bioreactors to a cell density of approximately $1 \times 10^5$ cells mL$^{-1}$ (see Supplementary Methods). UVR exposures were performed using Daavlin 305-12BB UVB and 350 12 UVA lamps. Experiments were performed in triplicate for both conditions (3× LC, 3× HC). On average both HC and LC replicates experienced individual contributions of 0.33 mW cm$^{-2}$ UVA (~67% of total UVR) and 0.16 mW cm$^{-2}$ of UVB (~33% of total UVR) for stage 2 totaling approximately 0.5 mW cm$^{-2}$ of UVR. Culture density was monitored by direct cell counts using a standard hemocytometer. Photosynthetic efficiency (maximal PSII quantum yield, $F_v/F_m$) was obtained from the maximal fluorescence ($F_m$) and variable fluorescence ($F_v$) using the Phyto-PAM (Pulse Amplitude Modulated) Phytoplankton Analyzer. Variable fluorescence was calculated from $F_m$ to $F_o$, where $F_o$ is the fluorescence yield when cells are dark acclimated. The pH of growth medium was determined spectrophotometrically using the indicatory dye *m*-cresol purple[61]. Total dissolved inorganic carbon (DIC) was measured using an Apollo SciTech (DIC analyzer) Model AS-C3 and Li-COR LI-7000 $CO_2$/$H_2O$ analyzer from filtered (0.2 μm) samples. Nitrate level was determined by Szechrome NAS colorimetric analysis as per the manufacturer's protocol (Polysciences Inc.). Specific growth rate $\mu$ (day$^{-1}$) was calculated from the linear regression of the natural log transformations of the cell concentrations vs. time (days) over the exponential phase of growth[7]. Growth curves were plotted by fitting growth data to a logistic model that includes the seeding estimates of maximum cell growth (i.e., carrying capacity), growth rate, and length of the lag phase[62] (please see the equation in the Supplementary Methods).

**RNA extraction, library construction, and sequencing**. Samples were harvested from early and late exponential phase cultures, during the light and dark phase (total four samples per stage, per replicate), and vacuum filtered onto 0.2 μm filters (GTTP 47 mm, Millipore) and immediately flash frozen in liquid nitrogen (Fig. 1, green triangles). Total RNA was extracted using the Spectrum Plant Total RNA Kit (Sigma Aldrich, STRN50). Genomic DNA contamination was removed with on-column DNase digestion (Sigma Aldrich, DNase 10). A total of 56 barcoded transcriptome libraries were prepared using the Illumina Truseq Stranded mRNA HT Library Prep Kit (cat# RS-122-2103). Libraries were pooled, denatured, and diluted according to the NextSeq 500 protocol. Paired-end sequencing of libraries was performed on the Illumina Nextseq 500 platform using a high-output 300 cycle v2 flowcell.

**Expression analysis**. Read files were cleaned with Trimmomatic version 0.33[63] and mapped with aligner STAR version 2.4.5a[64] to *T. pseudonana* genome version ASM14940v1.29 obtained from Ensembl[65–67]. Gene expression levels were quantified as fragments per kilobase of transcript per million mapped reads (FPKM)

**Fig. 4** A global transcriptome-derived map of physiological state transitions demonstrates that diatoms lose 'relational resilience' as they approach a point of no recovery. All 24 transcriptomes from the first stage (no UVR) of HC and LC conditions were used to identify sets of genes with distinct expression patterns for four physiological states associated with the diurnal cycle and growth phase (see Methods). **a** Illustrative representation of state descriptor expression distributions (boxplots) with distinct patterns in light (orange) vs. dark (green) or early (blue) vs. late (red) phases of growth and examples of how five hypothetical transcriptomes (markers) would coordinate in relation to the state descriptors. The expression distributions for all state descriptors from the stress test is in Supplementary Fig. 11 and listed in Supplementary Data 1. **b** A two-dimensional (2D) physiological state space was generated from the differential expression patterns of all state descriptors in stage 1 across both $CO_2$ conditions (see Methods). The five hypothetical transcriptomes (**a**) are mapped here into the 2D state space based on their expression patterns. For instance, transcriptomes in which state descriptors match an observed light/dark and early/late expression patterns in stage 1 would map to the associated corners of the state space (e.g., square and diamond markers), while a transcriptome in which the state descriptors have mixed expression patterns would localize to the "non-distinctive area" of the state space (e.g., both up and down triangles and the circle marker). The sampled transcriptomes of the stress test were mapped to the 2D state space for all stages of the LC (**c**) and HC (**d**) conditions. Replicate transcriptomes are shaded orange (light) or green (dark), based on when they were sampled; open markers represent samples from the early phase and filled markers represent late phases of growth (see key). Stages 1, 2, and 3 are represented by a circle, square, and triangle marker, respectively, and arrows illustrate temporal sequence of sampling in each stage and background ellipsoids highlight replicate groupings. Dashed circles represent 1.96 standard deviations from observed state positions in stage 1

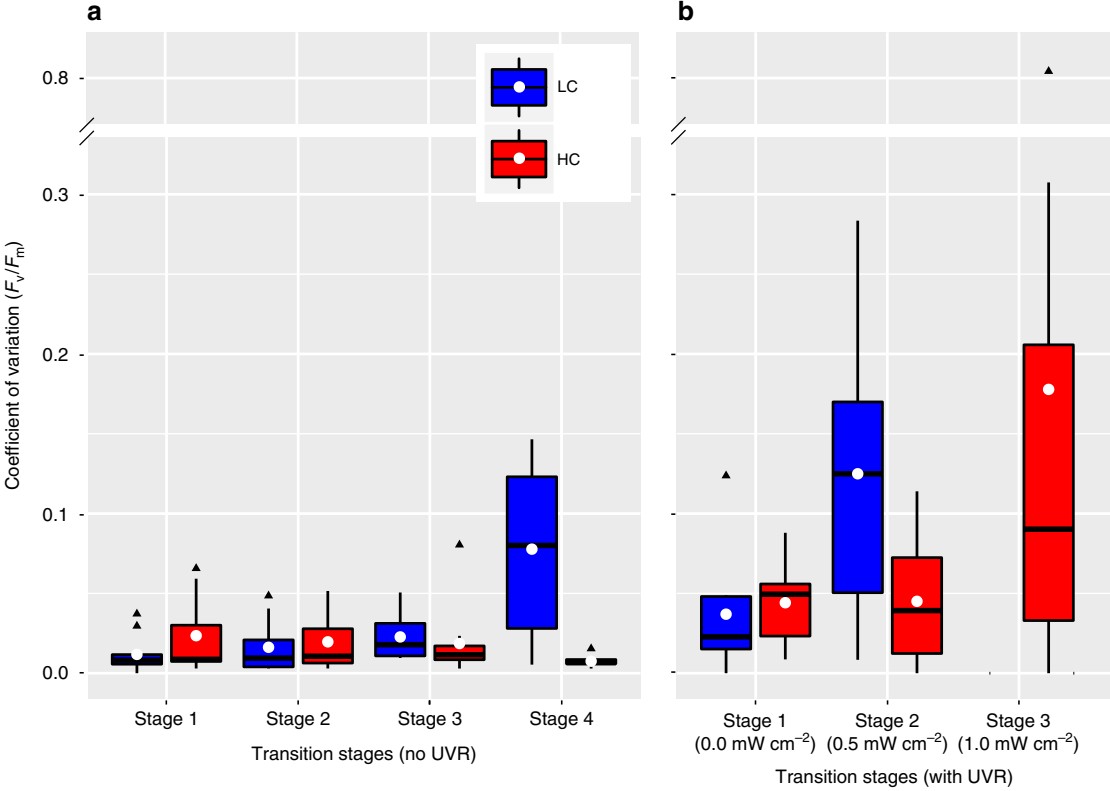

**Fig. 5** Increased coefficient of variation in photosynthetic efficiency may be an early warning sign of collapse. Coefficient of variation in $F_v/F_m$ of replicates over the growth cycle in each stage of the experiment for cultures grown in LC and HC conditions, without (**a**) and with (**b**) UVR during the stress test. A white dot represents the mean, median is indicated with a black horizontal line for each boxplot, and outliers are shown with black triangles

using Cufflinks[68]. Custom scripts were used to call different computational tools, each with specific parameter settings.

To test similarity between transcriptomes of biological replicates, we performed a Spearman's rank correlation coefficient analysis. To reduce noise, transcripts with very low expression (below 5 FPKM) were excluded from the analysis. The remaining transcripts resulted in a compendium of more than 8000 expressed genes consistently included at each comparison, thus providing a global view of replicate similarity.

**Transcriptome state analysis.** Twenty-four transcriptomes from stage 1 (LC and HC) were used to identify transcripts that could discriminate between (i) daytime and nighttime states, and (ii) early and late phase of growth. For a transcript to qualify as a light/dark or early/late state descriptor, its expression pattern had to (1) change in a statistically significant manner, and by twofold and (2) have at least one average standard deviation separation in $\log_{10}$-transformed FPKM across the two discriminated conditions. Altogether, 58 descriptor transcripts of light and dark states, and 218 descriptor transcripts of early and late growth phase (Supplementary Fig. 11 and Supplementary Data 1) were used to map a given transcriptome signature into a new 2D state space by computing a position score as follows:

$$s = \left[ s_x, s_y \right], \quad (1)$$

where $s$ is the overall positional coordinate, the abscissa $s_x$ refers to the diurnal cycle variable, and the ordinate $s_y$ refers to the growth phase variable.

Each dimension-specific coordinate was calculated separately, as a weighted average for each descriptor,

$$s_x = \sum_{i=1}^{n} \omega_i \lambda_i, \; s_y = \sum_{j=1}^{m} \omega_j \lambda_j, \quad (2)$$

such that weights for each dimension summed one:

$$\sum_{i=1}^{n} \omega_i = 1, \sum_{j=1}^{m} \omega_j = 1. \quad (3)$$

For a given transcriptome, we computed a position score $\lambda_i$ for each transcript $i$ from its expression value $\kappa_i$. Initially, during selection of state descriptors, we obtained for each transcript two expression distributions from stage 1, $\phi_\alpha$ and $\phi_\beta$

whose medians define the expected expression values $Q_2(\phi_\alpha)$ and $Q_2(\phi_\beta)$ for each physiological state, that is, light/dark and early/late for light cycle and growth phase phenotypic variables, respectively. Next, we first identified whether $\kappa_i$ was closer to $Q_2(\phi_\alpha)$ or $Q_2(\phi_\beta)$, using a piecewise function $\delta$,

$$\delta = \begin{cases} 1, & \text{if } \left| \kappa_i - Q_2(\phi_\alpha) \right| < \left| \kappa_i - Q_2(\phi_\beta) \right|, \\ -1, & \text{if } \left| \kappa_i - Q_2(\phi_\alpha) \right| > \left| \kappa_i - Q_2(\phi_\beta) \right|, \end{cases} \quad (4)$$

and also the relative position of $\phi_\alpha$ and $\phi_\beta$ using another piecewise function $\epsilon$,

$$\epsilon = \begin{cases} 1, & \text{if } Q_2(\phi_\alpha) < Q_2(\phi_\beta), \\ -1, & \text{if } Q_2(\phi_\alpha) > Q_2(\phi_\beta). \end{cases} \quad (5)$$

Then, if $\kappa_i$ was within the expression range of $\phi$, we linearly interpolated $\lambda_i$ from $\kappa_i$ as,

$$\lambda_i = \begin{cases} 1.5\delta + \frac{\epsilon |\kappa_i - Q_2(\phi)|}{2|\phi_{\min} - Q_2(\phi)|}, & \text{if } \kappa_i < Q_2(\phi), \\ 1.5\delta - \frac{\epsilon |\kappa_i - Q_2(\phi)|}{2|\phi_{\max} - Q_2(\phi)|}, & \text{if } \kappa_i > Q_2(\phi), \\ 1.5\delta, & \text{if } \kappa_i = Q_2(\phi). \end{cases} \quad (6)$$

In the case that $\kappa_i$ located at the gap $\theta$ within $\phi_\alpha$ and $\phi_\beta$, we linearly interpolated $\lambda_i$ as,

$$\lambda_i = \frac{2\epsilon(\theta_c - \kappa_i)}{\theta}, \quad (7)$$

where $\theta_c$ is the center of the gap $\theta$.

To assess the variability among culture replicates during each stage, we computed the Euclidean distance among all sample replicates for each time-point in the state space for LC and HC conditions. A Mann–Whitney $U$ test was used to assess the significance of distribution differences.

**Code availability.** The custom codes for growth curve, expression quantification, and transcriptome state analyses are available at https://github.com/adelomana/viridis.

**Data availability**. All sequences supporting the findings of this study are publicly available from the National Center for Biotechnology (NCBI) Sequence Read Archive (SRA), accession code PRJNA386016.

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

## Acknowledgements
We thank the National Science Foundation (Grants MCB-1316206 to M.V.O. and N.S.B.; DBI-1262637, DBI-1565166, MCB-1330912, and MCB-1616955 to N.S.B.) and the National Institutes of Health (Center award 2P50GM076547 to the Institute for Systems Biology). We would like to thank Jane E. Timberlake and Megan Schatz for their support of this research. We also acknowledge Pamela Troisch and Danielle Yi in the Molecular and Cell core at the Institute for Systems Biology for their help in next-generation sequencing services.

## Author contributions
J.J.V., E.V.A., M.V.O., and N.S.B. designed experiments. J.J.V. and A.L. performed all growth experiments, assays, RNA extraction, and preparation. A.L.G.d.L. performed RNA-Seq analysis and state space modeling. J.J.V. and A.L.G.d.L. performed growth and transcriptome analysis. J.J.V., A.L.G.d.L., A.L., E.V.A., M.V.O., and N.S.B. contributed to discussion of results. J.J.V., A.L.G.d.L., and N.S.B. wrote the manuscript.

## Additional information

**Competing interests:** The authors declare no competing interests.

