## [Peer Review File · Nature Communications]

Reviewers' comments:

Reviewer #1 (Remarks to the Author):

The authors provide an interesting study on diatom responses to UV stress under nitrogen limited, CO₂ enriched conditions. Diatoms are important players in the marine food web and there is a need to better understand their stress responses under future scenarios.

A replicated perturbation experiment was performed to test for UV effects on diatom growth rates and physiology, to ultimately probe the diatoms' resilience. Although use of transcriptome analysis to elucidate *T. pseudoana* stress responses is not new (see Crawford et al, 2012 and Ashworth et al 2013), this is still relatively uncommon in resilience studies, and a strong asset of the authors approach. Transcriptome analysis seems sound and well documented, and the paper reads well overall. However, I do have multiple concerns regarding culturing conditions and conclusions based on the experiments.

Firstly, although the experiment provided in the manuscript may show a higher resistance to UV stress in the HC treatment, I am not convinced the experiments show a clear loss/gain of resilience. Most importantly, the difference in cell density between controls (Epoch 1, LC (no UV) vs Epoch 1 LC - control (no UV), and Epoch 2 LC control (no UV)) shows a potential instability in culturing conditions that may bias growth rates under LC conditions. The max. density reached in Epoch 1 of the control is more than twice that of the impacted culture, even though both did not experience UV stress. Also, in the second epoch of the control, only half the former cell density is reached, without UV stress present. A significant difference between starting conditions is problematic, because it may affect the effect of the stressor. Especially, max. density is important because diatoms shade each other, which will shift critical UV-AB doses. The authors hardly mention variability in controls, and no information is available on actual UV doses experienced by the cells. Furthermore, the experimental set-up, including large dilution disturbances, caused crashes of the cultures after only 1 (LC) or 2-3 (HC) perturbations, making it difficult to infer resilience indicators.

Secondly, resistance to one type of stressor was measured for one diatom species, growing in isolation on one particular growth medium. Aren't the authors overgeneralizing their findings by stating in the title: 'Diatoms will be more resilient in an acidified ocean'? The finding that diatoms better tolerate UV stress under CO₂-acidified conditions is not new. See for example Wu et al 2014, as already referenced by the authors. The authors hypothesize reactions to other stressors to be similar, as more C can be allocated to stress mechanisms due to relieve of energy-intensive carbon concentrating mechanisms (which has been hypothesized previously), but whether this functions as expected when experiencing different types of stress remains to be tested. For example, others found no beneficial effect of elevated CO₂ on photosynthetic efficiency, and it has been found that increased PAR may lead to increased photoinhibition under elevated CO₂:

Wu, Y., Gao, K. and Riebesell, Ulf (2010) CO₂-induced seawater acidification affects physiological performance of the marine diatom *Phaeodactylum tricornutum* *Biogeosciences* (BG), 7 (9). pp. 2915-2923. DOI 10.5194/bg-7-2915-2010.

Many different definitions of resilience are currently in use. The type of resilience the authors state to observe seems to resemble 'engineering resilience' rather than 'ecological resilience'. Mention of 'relational resilience' needs to be clarified and referenced, and deviation from more common resilience definitions (i.e. Holling's) should be justified.

Minor comments

Line 74: Needs a reference. Here resistance is referred to as 'mechanisms to maintain a relevant physiological state in face of small perturbations' – which is resilience in Holling's definition.

Lines 99-102: Why did the authors choose to use batch cultures, and not a continuous culture set-up to create stable conditions?

Line 110: The cited papers indeed show variability between replicates, but these do not necessarily arise from 'bottlenecking'. In fact, the perturbations applied were much smaller than those presented in this manuscript.

Line 194: Equal amounts of UV-A and UV-B?

Line 124: should read Fig. 2 (?)

Line 125. I find it difficult to see evidence of 'critical slowing down' in this experiment. Authors refer to the increased lag phases of the cultures. Return to a base line of stable biomass would in my opinion make more sense. The perturbations may have been too large, and the increase in stress level too fast, to make reliable statements about slowed recovery (the crash occurs already after the first perturbation for LC, and after 2-3 for HC).

Figure 1. It would be interesting to plot the mean UV experienced by the cells in the medium, for example as was done for PAR in the photoinhibition study by Faassen et al 2015. (*Oikos* 124.12 (2015): 1617-1623).

Reviewer #2 (Remarks to the Author):

Reviewon: Diatoms will be more resilient in an acidified ocean by Valenzuela et al. submitted to Nature-communications

A) The manuscript by Valenzuela shows that the resiliency of the diatom *Thalassiosira pseudonana* to stress such as UV radiation is reduced under low CO₂ concentrations compared to the same species under enhanced CO₂ as we expect in a future ocean. The authors used standard growth and acclimation data as well as transcriptomic data to get to their conclusion.

B) To my knowledge the matrix with CO₂ and UVR is not that novel and data has been published for *T. pseudonana* by Sobrino et al 2008 (not cited in the ms). In Sobrino et al, the UVR partially counteracted the increased chl a based C-fixation observed under elevated CO₂. Similar experiments were done by Li et al 2012 or Wu et al 2012 (cited in the ms). However the mentioned manuscripts focused on photo-physiology and growth effects and did not use transcriptoms, which is the highlight of this ms.

C) The experimental setup was done appropriately and the data evaluation and statistics was performed well. The data are presented well, yet, as a person without transcriptomics background some of the figures / captions are hard to understand.

D) Statistics are done appropriately (to my knowledge) although statistics using n=3 (growth rates) are always questionable.

E) This very interesting data-set would – if we would be able to generalize – have large implications for ecosystem prediction and biogeochemical cycles.

Despite my general opinion, that this is a very nice experiment and data-set, I have some additional questions and concerns, which I detail below.

F) In my opinion, the authors should revise the background and add additional and needed information regarding known UVR response in Diatoms and how this can be linked to the energetic pathways of the cell> the authors might also consider to analyze their transcriptome dataset in more detail, including not only CCM regulation but also UVR regulation pathways. These new data could hence give better insight into the potential link between HC-CCM down-regulation and the UVR protection and overall resilience.

Please see my point-to-point critics below:

- Please change the header from "Diatoms" in general to "the diatom *Thalassiosira pseudonana*". Diatoms have large taxonomic differences, which leads to differences in CCM regulation, as well as in UVR responses. Since this manuscript only analyzed the response of one species it is overambitious to generalize the measured response.
 - The introduction is missing background on how diatoms respond to and regulate their metabolism to UVR stress. (e.g. what are protective strategies against UVR in diatoms? How much energy is assumed to be used for UV protection? Is there a link between UVR regulation and the CCM?) Since the authors state that energy reallocation could be responsible for the better resilience under HC, a paragraph about this would be necessary to better understand the big picture and the connections between pathways. This would also be necessary not only in the introduction but also in the results. The data should show a transcriptomic signal of up-regulated protection pathways during UVR stress. Please clarify and change
 - The authors state and show (Fig. S1) that the UVR applied is not lethal, yet it would be nice to know if there is cellular potential to acclimate to these new conditions and long it would take to get back to maximum growth rate.
 - The manuscript is partially based on the different response of 3 replicate cultures to LC/HC and UVR. Although it is standard to use triplicate cultures, a larger number of replicates would be better, especially when large deviations within the replicates are apparent, as happened in the experiment here. Did the authors find differences in the transcriptome within the replicates? It would be important to know what triggered these different responses.
 - It is unclear to me how the bottleneck was introduced in epoch III and IV. According to the authors the bottleneck was introduced after the cells reached late-logarithmic/stationary phase AND N-level was below 1 μ M. Yet in the UVR treated cultures cell density did not increase to values in which I would expect N limitation (especially since max carrying capacity was much higher in epoch I as well as in the control). Additionally, I did not see the data showing respective N concentrations. Do the authors mean "after the cells reached late-logarithmic/ stationary phase AND/OR N-level was below 1 μ M" with emphasis on the AND/OR. But even then I would need data to apprehend that indeed cells were N limited at the time of dilution.
 - Is there a possibility that the LC-UVR treatment in epoch III would have recovered after day 17 of the experiment? Two of the HC-UVR cultures recovered only around day 15 while the third did not recover, similar to the LC-UVR treatment. What if the cells just have needed a couple more days to start growing – this is related to an earlier remark of acclimation strategies of diatoms to UVR.
 - Since the authors base their assumptions of higher resilience under HC onto a potential down-regulation of the CCM, more information on regulation of certain CCM genes – not only a regulation of CA – should be shown. For example Diatoms have a multitude of CA, HCO₃⁻ and CO₂ transport systems, RubisCO regulation As shown by e.g. Rokitta et al 2014 in BMC Genomics (for the cocco. *E.hux*) one can nicely get more detailed understanding from the transcriptome regarding its C acquisition.
- G) Additional references regarding UVR regulation are needed.
- H) The ms is written clearly yet is missing some aspects as mentioned above.

Reviewer #3 (Remarks to the Author):

The manuscript by Valenzuela et al. describes the response of the diatom *Thalassiosira pseudonana* to UV exposure in cultures cultivated at low CO₂ (300 PPM) and high CO₂ (1000 PPM) conditions. Changes in growth rate and transcriptome shifts are quantified, with the goal of determining how resilient diatoms will be in a future acidified ocean. There has been considerable interest in predicting

how ocean organisms will respond to an acidified ocean, with largely detrimental impacts reported for calcifying organisms such as pteropods, corals, and coccolithophores. However not all studies report such detrimental impacts, and this variability makes predicting the biological response to ocean acidification (OA) a continued challenge. Some of the confusion surrounding conflicting reports on the marine biological responses to OA has been generated by technical issues in regulating carbonate chemistry, yet a clear and mechanistic understanding of how different functional types of organisms will respond has not yet emerged. Considering this, the study reported in this manuscript addresses a highly relevant issue.

How diatoms will respond to an acidified ocean is an outstanding question in the field. It is recognized that the growth rate of diatoms is limited by the supply of CO₂ (Riebesell et al. 1993 Nature 361: 249-251). However this is a complex issue and contradictory results have been published as well (reviewed in Li and Capbell 2013, PLoS ONE). Consequently, it is unsurprising that diatoms may benefit from a future scenario in which OA increases the pCO₂ supply. This study is novel in its attempt to quantify "resilience" of diatoms cultured under low and high CO₂, thereby providing some mechanistic insight into how diatoms may benefit in a proposed future ocean scenario. However, this study is not successful in achieving this objective, with the most fundamental issue being a flawed experimental design that does not actually test the effect of UV on *T. pseudonana* cultured at different CO₂ conditions, as it sets out to do.

Major points:

The assertive and colorful title overstates conclusions which are quickly undermined by the data presented.

The hypothesis that Valenzuela et al. put forth is that energetic savings from reduced CCM expenditures in diatoms at high CO₂ will improve resilience. The literature they cite discusses the CCM in an evolutionarily distinct diatom *Phaeodactylum tricornutum*, and it is a stretch to make similar assumptions about how the CCM operates in the diatom studied here, *T. pseudonana*. In fact the energetic cost model for the biophysical CCM function cited is based on the occurrence of a pyrenoid carbonic anhydrase which is absent in *T. pseudonana* (Tachibana et al. 2011 Photosynthesis Res). Although *T. pseudonana* does have CAs located in the stroma this is an important distinction. It is unclear if the CCM that *T. pseudonana* possesses even requires energy or if it is a passive system. Therefore, this is an unreasonable leap.

The primary issue with this study is the experimental design and controls. Growth curves don't replicate between the UV challenged and control experiments at both LC and HC conditions during Epoch I, which raises questions and concerns. According to the way the experimental design and results are reported, in Epoch I the pairs of curves plotted in Fig. 1a and 1c in and 1b and 1d should represent the same conditions. However the maximum cell densities in the controls (c and d) appear 2x higher (reached on shorter timescales) than the UV experiments (a and b) (yet in Epoch I where this significant difference arises there has not yet been any UV exposure)? Further what is limiting the growth of LC cultures at the end of Epoch I in the UV (which has not yet been treated with UV) and the non UV? If it's N-limitation, as the authors state, then why are neither the control experiment nor the HC cultures similarly N-limited? Most problematic is that authors attribute the impaired growth observed in Epoch II in LC relative to HC to the UV exposure, whereas the simpler (and more likely) explanation is that less healthy cells were transferred between epochs from LC than those in HC. What is likely observed in epoch II is an increase in the length of lag phase in LC which cannot be assigned to the UV exposure treatments. The proper control to test the UV effect would have transferred inocula from the end of each epoch into UV-exposed and non UV-exposed batches, and observed the difference caused by UV alone independent of other variables which influence growth characteristics.

This is a major issue with the experimental design that prevents decoupling the effect of LC/HC from UV and therefore the conclusions drawn in the manuscript cannot be adequately evaluated. For example in Epoch II under LC it is critical that we see the impact of the same inoculum exposed to UV and not exposed to UV. The Epoch II control used for comparison to the Epoch II UV treatments is problematic because it is not the same inoculum and because the Epoch I cultures in the control and UV (although not yet exposed to UV) are clearly very different. In addition to major differences in yield these cultures appear to have been sampled at different phases of growth (the UV treatment cultures (Epoch I) were clearly sampled in stationary phase but this doesn't appear to be the case the control series). Obviously there are differences between LC and HC in the UV that do not appear as apparent in the controls but it is difficult to decouple the effect of LC and HC from UV without exposure of each LC and HC Epoch inoculum to UV and non UV conditions.

Because the experimental design is plagued with aforementioned issues, the interpretation of state shifts obtained by transcriptome sequencing is compromised & difficult to interpret. It does seem apparent and convincing that there is a quantifiable loss of "relational resilience" observed in LC cultures, however as previously discussed, the UV stress effect cannot be decoupled from normal batch culture resource limitation dynamics. It's unclear why quantifying state transitions and relational resilience is better than other more well accepted metrics for fitness or survivability such as cell viability (i.e. with SYTOX Green) or LD50. It would however been informative to establish linkages between such physiological assays and the transcriptomics data.

Finally, this manuscript doesn't cite appropriate literature. For example, Shi et al. 2010 Effect of Ocean Acidification on Iron Availability in Marine Phytoplankton (Science 327: 676-679) is one of the most highly cited and relevant papers on the subject, but is not referenced or discussed in this manuscript. The synergistic environmental changes diatoms will experience in a future acidified ocean should be considered, particularly in a study which claims to predict how these organisms will respond. The idea that diatoms are predicted to face difficulties regarding Fe availability in response to OA conflicts with the claim presented here that diatoms will be more resilient under OA conditions. Studies evaluating the impact of OA on diatoms need to consider this (or at least mention it). This manuscript also neglects to mention seminal studies from the coccolithophore literature, even in order to contrast the response of diatoms to other phytoplankton which have been more well studied in an OA context. This is an important discussion point to put their results in a broader context.

Minor points:

Terminology applied to the batch culture experiments is grandiose. Calling growth cycles "epochs" is misleading. It connotes evolutionary timescales and conjures up the sense that within each growth cycle there is genetic adaptation. Can they be termed cycles? Similarly, calling inoculum transfer a "bottleneck" (line 104-105) also implies some significance evolutionarily which is unnecessary and obfuscatory. Or this needs to be much better justified or explained is there is some specific point being made.

A description of curve fitting to growth data is not provided and in some cases it looks like S-curves were forced to fit which misleads the reader into thinking that stationary phase was reached. There are also no error bars on cell counts (hemocytometer counts are notoriously variable and models aimed at fitting growth curves should account for this).

Authors state that N is depleted below the detection limit (lines 103-104) and should include these data.

Summary of response

We were very encouraged by all three reviewers' enthusiasm for the approach and findings on how future ocean conditions will alter resilience of *Thalassiosira pseudonana*. We appreciate their thoughtful comments, suggestions and critiques, which have been addressed in their entirety through additional experimentation, data analyses, and extensive revisions to strengthen the main conclusion of this study. Here's a brief summary and few highlights of what we have done:

1. **Stability and carrying capacity of cultures.** We conducted an additional control experiment to assess whether *T. pseudonana* cultures might have collapsed because an aliquot of an unhealthy culture was transferred to the next stage. Results in **Figure R11** demonstrate that cell cultures are in fact stable at the time of transfer, and that they recover without effect in the absence of UV radiation (UVR). Analysis of growth characteristics across all experiments also ruled out carrying capacity as a variable that might have played a role in the collapse of cultures (**Fig R2**).
2. **Evidence for N-limitation.** We have measured N levels in all cultures, through all stages of the experiment, and have included N drawdown profiles to show that all cultures were indeed N-limited (**Fig R3**). We have also provided additional evidence through transcriptome analysis that N-limitation led to induction of N-metabolism genes (**Fig R8**).
3. **Measured UVR experienced inside reactors.** We have measured UVR experienced by cells inside reactors to demonstrate that there was no difference across HC and LC conditions, difference across cultures with different Fv/Fm, or any self-shading in cultures with higher cell density (**Fig R4**).
4. **Reproducibility of transcriptomics.** We have reanalyzed the transcriptomics data to demonstrate that there was indeed high degree of reproducibility across replicates and related growth conditions, except in cultures that were unstable and on the brink of collapse (**Fig R7**).

We have also expanded discussion on some topics such as UVR, and provided more information and details on experimental design and interpretations. All of the revisions are tracked and correlated to specific reviewer critiques. Below we provide a point-by-point response to each reviewer critique.

Reviewer #1:

The authors provide an interesting study on diatom responses to UV stress under nitrogen limited, CO2 enriched conditions. Diatoms are important players in the marine food web and there is a need to better understand their stress responses under future scenarios.

*A replicated perturbation experiment was performed to test for UV effects on diatom growth rates and physiology, to ultimately probe the diatoms' resilience. Although use of transcriptome analysis to elucidate *T. pseudonana* stress responses is not new (see Crawford et al, 2012 and Ashworth et al 2013), this is still relatively uncommon in*

resilience studies, and a strong asset of the authors approach. Transcriptome analysis seems sound and well documented, and the paper reads well overall. However, I do have multiple concerns regarding culturing conditions and conclusions based on the experiments.

CRITIQUE 1: Firstly, although the experiment provided in the manuscript may show a higher resistance to UV stress in the HC treatment, I am not convinced the experiments show a clear loss/gain of resilience.

RESPONSE: It is incorrect to interpret from our results that *T. pseudonana* growing in HC conditions has higher resistance to UVR stress, for one simple fact - cultures in both HC and LC conditions collapsed at a very low and minimally growth sub-inhibitory dosage of UVR (**Fig R1**). In other words, the cultures did not succumb to the lethal effect of UVR, and hence the experiment design did not quantify resistance. On the contrary, the experiment design quantified the ability of *T. pseudonana* to recover from incrementally higher amounts of low dose UVR –an established systems property called “resilience”¹⁻⁷. The uniqueness of our experiment design comes from imposing additional ecologically relevant constraints, such as day/night cycles and changes in N availability, in order to mimic the natural environment of *T. pseudonana*. This complex experiment design probes the ability of *T. pseudonana* to effectively manage trade-offs in allocating resources towards UVR stress response and recovery, while having to deal with

complex, routine changes that require resource-intensive, large-scale physiological re-adjustments. A small increase in resource availability under HC-conditions shifted the balance in this trade-off, and manifested in

increased resilience, i.e., a shift in the bifurcation point between recovery and collapse. We have clarified this point in the manuscript (page 4, lines 84-87) and (**Fig 1a and b** in manuscript).

CRITIQUE 2: Most importantly, the difference in cell density between controls (Epoch 1, LC (no UV) vs Epoch 1 LC- control (no UV), and Epoch 2 LC control (no UV)) shows a potential instability in culturing conditions that may bias growth rates under LC conditions. The max. density reached in Epoch 1 of the control is more than twice that of the impacted culture, even though both did not experience UV stress. Also, in the second epoch of the control, only half the former cell density is reached, without UV stress present.

RESPONSE: In order to investigate whether carrying capacity might have influenced whether and when the cultures collapsed, we compiled data from up to 13 different

experiments (8 LC and 5 HC) done at three different times with subtle differences in initial media conditions and inocula.

Fig. R2. Carrying capacity of replicate cultures across different stages of the stress test experiment. Bean plots show carrying capacities of three sets ($n=8$) of replicate experiments for LC conditions and two sets ($n=5$) for HC conditions. Solid black lines indicate mean of the bean plot, ticks represent one-dimensional plot of the carrying capacity, and the y-axis is log scaled.

The compiled results demonstrate that there was no significant difference in the overall distribution of initial carrying capacities between HC and LC cultures in the first epoch (hereon labeled 'stage'). In stage 2 when cultures experience a small dose of UVR (0.5 mW/cm^2), all LC cultures are affected and have lower carrying capacities. By contrast, HC cultures appear to be at a bifurcation point with an even distribution between cultures that are affected and those that are not.

To understand how differences in initial conditions might have impacted the outcome of specific cultures, we examined nitrate consumption profiles across cultures from the stress test and non-UVR transition experiment with the biggest differences (lower end: ~ 60 -

Fig R3 Nitrate consumption profiles for LC and HC cultures, with and without UVR stress.

$70 \mu\text{M}$ nitrate, higher end: 90 - $110 \mu\text{M}$), both with and without stress, respectively. Results of this analysis also confirmed that carrying capacity has no effect on the phenomenon. Most significantly, all cultures had consistent profiles for drawdown ($< 1 \mu\text{M}$) of nitrate, regardless of initial nitrate levels, CO_2 levels, carrying capacities, or stage. The only cultures that showed a deviation in the nitrate drawdown profiles were ones that collapsed. While it is clear from these results that not all of the nitrogen (N) is

going towards biomass production, further studies would be necessary to track how it is being utilized for stress management, such as through production of mycosporine-like amino acids (MAAs)⁸, or generating N reserves^{9,10} (**Fig R3; Fig S1** in the manuscript).

Based on the results presented here we can conclude that carrying capacity had no effect on when cultures collapsed.

CRITIQUE 3: *Especially, max density is important because diatoms shade each other, which will shift critical UV-AB doses. The authors hardly mention variability in controls, and no information is available on actual UV doses experienced by the cells.*

RESPONSE: We have measured the dosage of UVR within a reactor, and observed no difference in actual amount of UVR experienced by cells due to phase of growth, cell density, CO₂ levels, or cell state with respect to Fv/Fm (see **response to critique 16** below). While it is difficult to measure or quantify the effect of self-shading by diatoms, the variability appears to be low or a non-factor, based on little correlation between cell density and collapse dynamics of replicate cultures across stages. For instance, in stage 2 replicate C in HC conditions must have experienced higher per cell UV-AB irradiation relative to replicate B, which had achieved higher cell density. Yet replicate C recovered in stage 3 whereas replicate B collapsed (see **Fig 1**). If self-shading were a factor than the outcome would have been the exact opposite. The response to critique 2 above also explains why carrying capacity was not a factor that influenced collapse dynamics.

CRITIQUE 4: *Furthermore, the experimental set-up, including large dilution disturbances, caused crashes of the cultures after only 1 (LC) or 2-3 (HC) perturbations, making it difficult to infer resilience indicators.*

RESPONSE: The experimental set-up and large dilutions did not cause the cultures to collapse; we know this because cultures that were treated identically but not subjected to UVR did not collapse (See **Fig 1c and d** in the manuscript). This experiment has been repeated numerous times in addition to what we have reported in the manuscript, and the results are *highly reproducible* in terms of the time-frame when (number of days, stage, and UVR dosage) the cultures collapse in LC and HC conditions. In other words, the cultures collapse only when they experience a very small amount of UVR for one hour each day at noon. But we do agree that smaller step increases in UVR irradiation will help to better delineate resilience indicators as has been done previously^{2,3,11}. However, it is important to highlight a key difference between the design of these previous studies and the study we have conducted.

Our primary objective was to develop a framework to quantify change in resilience of *T. pseudonana* in response to a projected ecologically relevant change in CO₂, AND conduct temporal assessment of system-wide changes in gene expression to understand underlying mechanisms. In order to do this, we subjected diatoms to conditions that simulated complex, naturally occurring environmental changes in multiple factors such as day/night cycles, N availability, and UVR exposure. The thinking here was that a diatom like any other biological system has to multitask in order to deal with numerous environmental challenges simultaneously. The multitasking requires *T. pseudonana* to manage trade-offs in allocating resources to different ecologically relevant tasks, such as shifting between daytime and nighttime physiology, and growing in N-replete and deplete conditions. We demonstrated that in absence of UVR stress the diatoms indefinitely sustain growth in this dynamically changing environment. However,

the diatom cultures progress towards collapse in a very characteristic and highly reproducible timeframe, when they are required to do all of this multi-tasking while having to respond and recover from a small amount of UVR stress experienced just for an hour each day at noon time. This was a very complex experiment and one that required enormous optimization and attention to detail (a point appreciated by reviewer 2).

This experiment design allowed us to quantify resilience in terms of the number of day/night cycles, and stages over which *T. pseudonana* can recover from UVR stress to sustain growth. A subtle change such as increased CO₂ availability shifted this equilibrium and improved resilience of the diatom by enhancing its ability to sustain this taxing growth schedule over a longer number of days and stages (~18 day/night cycles and three stages in HC conditions compared to ~9 day/night cycles and two stages in LC conditions). The temporal assessment of transcriptome changes uncovered how the internal transcriptome state of the diatom became uncorrelated with changes in day/night cycles and N availability. Previous studies would not have uncovered this insight because they did not take into account such complex environmental changes; and that is because they utilized a chemostat to grow cultures under constant photoautotrophic conditions^{3,12}. Again, that being said we do plan to follow up our findings with a more elaborate experiment of smaller step increases in UVR so we can infer indicators of resilience at greater resolution.

CRITIQUE 5: *Secondly, resistance to one type of stressor was measured for one diatom species, growing in isolation on one particular growth medium. Aren't the authors overgeneralizing their findings by stating in the title: 'Diatoms will be more resilient in an acidified ocean'?*

RESPONSE: We submit that we might have overly generalized our findings to all diatoms. We have changed the title to "**The marine diatom *Thalassiosira pseudonana* is more resilient in an acidified ocean**".

CRITIQUE 6: *The finding that diatoms better tolerate UV stress under CO₂-acidified conditions is not new. See for example Wu et al 2014, as already referenced by the authors.*

RESPONSE: Our study was **not** intended to investigate UVR tolerance of *T. pseudonana*, which we agree has been done previously and appropriately cited in our manuscript. Our primary objective was to develop a framework to quantify change in resilience of *T. pseudonana* in response to a projected ecologically relevant change in CO₂, AND conduct temporal assessment of system-wide changes in gene expression to understand underlying mechanisms.

CRITIQUE 7: *The authors hypothesize reactions to other stressors to be similar, as more C can be allocated to stress mechanisms due to relieve of energy-intensive carbon concentrating mechanisms (which has been hypothesized previously), but whether this functions as expected when experiencing different types of stress remains to be tested. For example, others found no beneficial effect of elevated CO₂ on photosynthetic*

*efficiency, and it has been found that increased PAR may lead to increased photoinhibition under elevated CO₂: (Wu, Y., Gao, K. and Riebesell, Ulf (2010) CO₂-induced seawater acidification affects physiological performance of the marine diatom *Phaeodactylum tricornutum* Biogeosciences (BG), 7 (9). pp. 2915-2923. DOI 10.5194/bg-7-2915-2010)*

RESPONSE: Indeed, we have not tested whether our results generalize to other diatoms. However, based on their studies on *P. tricornutum*, Wu *et al.* suggest that CCM may act as a sink for excessive energy resulting from increased activity of PSII. This phenomenon is very likely generalizable to *T. pseudonana*. In fact, their findings support our hypothesis that down regulation of CCMs in HC conditions makes available excess energy that could be reallocated for managing stress. The important point to note is that Wu *et al.* assessed the effect of HC and LC conditions on productivity (photosynthetic C-fixation and growth rate) of *P. tricornutum* in response to different light intensities. They did not quantify resilience, which requires a different experiment design like the one we have used.

CRITIQUE 8: *Many different definitions of resilience are currently in use. The type of resilience the authors state to observe seems to resemble ‘engineering resilience’ rather than ‘ecological resilience’... and deviation from more common resilience definitions (i.e. Holling’s) should be justified.*

RESPONSE: Holling’s definition is indeed most relevant to our work, i.e. “a measure of the ability of these systems to absorb changes of state variables, driving variables, and parameters, and still persist”¹³. We have made edits on page 3, lines 73-74.

CRITIQUE 9: *Mention of ‘relational resilience’ needs to be clarified and referenced*

RESPONSE: The definition of “relational resilience” has been clarified and appropriate references are now cited^{4,14} - page 9, lines 198-200.

Minor comments

CRITIQUE 10: *Line 74: Needs a reference. Here resistance is referred to as ‘mechanisms to maintain a relevant physiological state in face of small perturbations’ – which is resilience in Holling’s definition.*

RESPONSE: This has been resolved.

CRITIQUE 11: *Lines 99-102: Why did the authors choose to use batch cultures, and not a continuous culture set-up to create stable conditions?*

RESPONSE: We made a deliberate decision to use batch culture because we wanted to create conditions in which the diatom was required to transit through four physiologic states with respect to light/dark and nitrate-replete/nitrate-limited conditions. See our more detailed response to **critique 4** to understand why this was important.

CRITIQUE 12: *Line 110: The cited papers indeed show variability between replicates,*

but these do not necessarily arise from ‘bottlenecking’. In fact, the perturbations applied were much smaller than those presented in this manuscript.

RESPONSE: We agree with the reviewer that variability does not arise because of bottlenecking. Variability arises from the perturbation imposed on dynamically transitioning population. The bottlenecking only “amplifies” the variability as denser populations may mask the loss of resilience. Additionally, we have changed the term “bottlenecking” to dilution as recommended by reviewer 3.

CRITIQUE 13: *Line 194: Equal amounts of UV-A and UV-B?*

RESPONSE: On average both HC and LC replicates experienced 0.33 mW/cm² of UV-A and 0.16 mW/cm² of UV-B during stage 2, totaling approximately 0.5 mW/cm² of UVR. We have added this information to the Methods on page 11, lines 242-244.

CRITIQUE 14: *Line 124: should read Fig. 2 (?)*

RESPONSE: Done –Correct.

CRITIQUE 15: *Line 125. I find it difficult to see evidence of ‘critical slowing down’ in this experiment. Authors refer to the increased lag phases of the cultures. Return to a base line of stable biomass would in my opinion make more sense. The perturbations may have been too large, and the increase in stress level too fast, to make reliable statements about slowed recovery (the crash occurs already after the first perturbation for LC, and after 2-3 for HC).*

RESPONSE: We agree that the current experiment design does not provide sufficient number of data points to support a claim of critical slowing down. The manuscript has been appropriately revised.

CRITIQUE 16: *Figure 1. It would be interesting to plot the mean UV experienced by the cells in the medium, for example as was done for PAR in the photoinhibition study by Faassen et al 2015. (Oikos 124.12 (2015): 1617-1623).*

RESPONSE: We have generated a plot of the mean UVR experienced by cells in the medium as in Faassen *et al.* 2015 and generated the requested plot by performing an additional experiment (**Fig R4**). However, we found it not to be as informative as it is in Faassen *et al.* because the design and objectives of our experiment was quite different. Faassen *et al.* probed a larger number of light intensities (346-1134 μmol photons m⁻² s⁻¹) over five months with small increases of 34 μmol photons m⁻² s⁻¹. In contrast, the original stress test experiment maintained a UVR dosage over four successive growth cycles at 0, 0.5, 1.0, and 1.5 mW/cm², respectively. However, we feel Figure 1 and 2 in the manuscript does already capture the effect of UVR on diatoms in HC and LC.

Mean Light Intensity (light experienced by cells) $I_{\text{mean}} = \frac{I_{\text{in}} - I_{\text{out}}}{\ln(I_{\text{in}}) - \ln(I_{\text{out}})}$
Equation R1. Mean light intensity as described by Faassen et al. 2015. Substituted UVR (mW/cm²) with light (μmol photons m⁻² s⁻¹)

Figure R4. Scatter plot of mean UV (UVR, UV-A, UV-B intensity experienced by cells inside photo-bioreactors. A) Growth dynamics of *Thalassiosira pseudonana* cultures in HC and LC conditions with nitrate limitation and a 12:12hr light/dark cycle. Arrows indicate time points when UVA, UVB, and UVR measurements were made. B) UVR exposures of 0.5 and 1.0 mW/cm² were applied and the mean UVR intensity (mW/cm²) was calculated as in Faassen *et al.* 2015. Each mean UVR intensity point from duplicate cultures is plotted against the corresponding photosynthetic efficiency (Fv/Fm, see Methods). Size of points are proportional to cell density (cells/mL) –see key.

Reviewer #2:

*The manuscript by Valenzuela shows that the resiliency of the diatom *Thalassiosira pseudonana* to stress such as UV radiation is reduced under low CO₂ concentrations compared to the same species under enhanced CO₂ as we expect in a future ocean. The authors used standard growth and acclimation data as well as transcriptomic data to get to their conclusion.*

CRITIQUE 1: *To my knowledge the matrix with CO₂ and UVR is not that novel and data has been published for *T. pseudonana* by Sobrino et al 2008 (not cited in the ms).*

RESPONSE: We apologize for this important omission, and now discuss and cite Sobrino's findings on effect of elevated CO₂ and UVR on *T. pseudonana*. We have also included more background on diatom UVR responses as per reviewer 2's request (see page 5, lines 109-110).

CRITIQUE 2: *The experimental setup was done appropriately and the data evaluation and statistics was performed well. The data are presented well, yet, as a person without transcriptomics background some of the figures / captions are hard to understand.*

RESPONSE: We recognize that this work covers diverse topics and techniques and we have done our best to make the information accessible to a broad audience. We have made minor adjustments to figure 3 and 4 (including headings) to make it more understandable for the reader (see page 8, lines 169-171, 178, and 179).

CRITIQUE 3: *Statistics are done appropriately (to my knowledge) although statistics using n=3 (growth rates) are always questionable. This very interesting data-set would – if we would be able to generalize – have large implications for ecosystem prediction and biogeochemical cycles.*

RESPONSE: We did repeat the experiment numerous times to ensure reproducibility of the collapse phenomenon and when it occurred (see **Fig R2**). Instead of doing RNA-seq on numerous replicates for just two time points (with and without treatment) we performed RNA-seq in triplicate, over a time course, across multiple day/night cycles, and growth phases for both HC and LC conditions, to generate a compendium of 56 genome-wide transcriptome profiles. The reproducibility of this data is presented in response to reviewer 2's critique 11 below.

CRITIQUE 4: *Despite my general opinion, that this is a very nice experiment and data-set, I have some additional questions and concerns, which I detail below. In my opinion, the authors should revise the background and add additional and needed information regarding known UVR response in Diatoms and how this can be linked to the energetic pathways of the cell.*

RESPONSE: We have added additional text (see page 5, lines 104-112).

CRITIQUE 5: *The authors might also consider to analyze their transcriptome dataset in more detail, including not only CCM regulation but also UVR regulation pathways. These*

new data could hence give better insight into the potential link between HC-CCM down-regulation and the UVR protection and overall resilience.

RESPONSE: We have performed additional analysis as requested by reviewer 2. The new analysis shows moderate upregulation of few known UVR and oxidative stress response-related genes^{15,16} with minimal difference across HC and LC conditions (**Fig R5**). However, the trends are not dramatic and do not add valuable insight that is worthy of inclusion in the manuscript. This might be attributable to the time course of sampling, which was not suited for assessing UVR response, as much of the differential regulation is likely to occur immediately post-UVR exposure. Our objective was to study resilience and not response to UVR *per se*. A higher time-point resolution experiment that sampled RNA immediately post UVR stress, would provide greater insight. This certainly offers an opportunity for future experiments to directly link down-regulation of CCM under HC conditions and the UVR stress response.

CRITIQUE 6: Please change the header from “Diatoms” in general to “ the diatom *Thalassiosira pseudonana*”. Diatoms have large taxonomic differences, which leads to differences in CCM regulation, as well as in UVR responses. Since this manuscript only analyzed the response of one species it is overambitious to generalize the measured response.

RESPONSE: We have changed the manuscript title to “**The marine diatom *Thalassiosira pseudonana* is more resilient in an acidified ocean.**”

CRITIQUE 7: *The introduction is missing background on how diatoms respond to and regulate their metabolism to UVR stress. (e.g. what are protective strategies against UVR in diatoms?)*

RESPONSE: We have added additional text to the manuscript (see page 5, lines 101-109).

CRITIQUE 8: *How much energy is assumed to be used for UV protection? Is there a link between UVR regulation and the CCM?*

RESPONSE: In our prior work we discovered that a major consequence of growth in HC conditions is the down regulation of CCMs (Hennon *et al.* 2015). This phenomenon has also been reported by others^{12,17-19}. Moreover, as pointed out by reviewer 1, the study by Wu *et al.* has also uncovered that CCM may act as a sink for excessive energy resulting from increased activity of PSII. The final observation comes from our study that cultures grown in HC conditions can sustain a higher growth sub-inhibitory dose of UVR while dealing with day/night cycles and changes in N-availability. Together, all these studies provide evidence for a link between UVR response and CCM. However, it is unclear if regulation of CCM and UVR are mechanistically linked or if the link is indirect. Our future plan is to extend the diatom gene regulatory network model (<http://networks.systemsbiology.net/diatom-portal/>)²⁰ that we have previously constructed through detailed transcriptome analysis of the UVR response in HC and LC conditions. Specifically, the transcriptome datasets will have to be generated over a time course like the one conducted by Wu *et al.* 2014 (see Fig. 5 from Wu *et al.* 2014 –reproduced here as **Fig R6** for benefit of the reviewer)²¹.

Finally, how much energy is devoted to UVR protection will depend on many factors including (i) exposure time and UVR dosage; (ii) how the organism repairs cellular damage to DNA and protein; (iii) whether and how much it produces protective pigments including mycosporine-like amino acids; and (iv) the degree to which these systems are engaged in the protection, repair and recovery process. Alternatively, one could use microcalorimetry to quantify amount of heat dissipated during recovery from UVR stress²² as a proxy for total energy devoted to UVR protection. From a photosynthetic standpoint NPQ (non-photochemical quenching) could be another way to link energy expenditures to mechanistic response^{21,23}. Indeed, the experimentation

framework we have developed is a step in the direction towards addressing these gaps, but they fall outside the scope of the current manuscript.

CRITIQUE 9: *Since the authors state that energy reallocation could be responsible for the better resilience under HC, a paragraph about this would be necessary to better understand the big picture and the connections between pathways. This would also be necessary not only in the introduction but also in the results. The data should show a transcriptomic signal of up-regulated protection pathways during UVR stress. Please clarify and change*

RESPONSE: We have analyzed expression changes and observed few UVR and oxidative stress response genes have higher expression in HC conditions (**Fig R5**). However, the upregulation pattern is only suggestive and will require a more in depth transcriptome survey immediately post-UVR exposure. However, we have addressed the reviewer's request by adding language to the manuscript that discusses our findings in the context of prior work by us and others on the relationship between UVR regulation and CCM (see first part of response to previous critique).

CRITIQUE 10: *The authors state and show (Fig. S1) that the UVR applied is not lethal, yet it would be nice to know if there is cellular potential to acclimate to these new conditions and long it would take to get back to maximum growth rate.*

RESPONSE: Wu *et. al* have performed the experiment requested by the reviewer and discovered that cultures grown under HC conditions have better recovery rates of photosystem II (PSII) relative to LC cultures, after 60 minutes of UVR. While HC cultures reached maximum photochemical yield within an hour post-UVR exposure, LC grown cultures achieved maximum photochemical yield in two hours, at which point both sets of cultures had recovered. We expect to see similar dynamics of recovery from sub-inhibitory dose of UVR, and that the observations would be consistent with our finding that HC cultures have more resources available to deal with stressful conditions. See also response to **critique 13**.

CRITIQUE 11: *The manuscript is partially based on the different response of 3 replicate cultures to LC/HC and UVR. Although it is standard to use triplicate cultures, a larger number of replicates would be better, especially when large deviations within the replicates are apparent, as happened in the experiment here. Did the authors find differences in the transcriptome within the replicates? It would be important to know what triggered these different responses.*

RESPONSE: We performed a Spearman's rank correlation analysis that shows transcriptomes from replicate cultures were highly similar in stage 1 across all time-points. Importantly, all day-time, night-time, early and late growth phase transcriptomes clustered into distinct groups –recapitulating four previously reported growth states of *T. pseudonana* in Ashworth *et al.* 2013. (**Fig R7; included as Fig 3 in manuscript, see page 7 and 8, lines 162-169 and page 12 lines 275-279**). As expected, there is relatively lower similarity across replicate transcriptomes from LC conditions in stage 2, and HC conditions in stage 3. This result is consistent with variable behaviors of

replicate cultures in later stages, especially prior to collapse. The variability across replicates does not reveal any telltale signs of specific mechanisms that have gone awry; rather this appears to be a consequence of global dysregulation of transcription. This is actually not surprising because it is well established that increased variability is a characteristic property of complex systems on the brink of collapse^{2,3,24}. Please also see response to critique 3.

CRITIQUE 12: *It is unclear to me how the bottleneck was introduced in epoch III and IV. According to the authors the bottleneck was introduced after the cells reached late-logarithmic/stationary phase AND N-level was below 1uM. Yet in the UVR treated cultures cell density did not increase to values in which I would expect N limitation (especially since max carrying capacity was much higher in epoch I as well as in the control). Additionally, I did not see the data showing respective N concentrations. Do the authors mean “after the cells reached late-logarithmic/ stationary phase AND/OR N-level was below 1uM” with emphasis on the AND/OR. But even then I would need data to apprehend that indeed cells were N limited at the time of dilution.*

RESPONSE: It is true that the number of cells transferred to the new condition with a final density of 100,000 cells/mL does not introduce a bottleneck in the classical sense. Reviewer 3 had the same concern, so we have revised the language to say that the culture was “diluted” by transferring a small aliquot to fresh medium with replete nitrate. But the description of when cultures were diluted was accurate –they were indeed transferred when nitrate levels were limiting (below detection or <1μM) and their photosynthetic efficiency (Fv/Fm) had decreased to ~0.30. To clarify this point we have included nitrate depletion curves (see **Fig R3**, and response to reviewer 1, critique 2).

The cultures had lower carrying capacity even though they consumed all available nitrate. The N-starved state of cultures was also confirmed by the observation that cultures in the late growth phase (just prior to transfer) significantly up regulated four genes of the nitrate assimilation module

(Thaps_hclust_0360: nitrate transporter (NRT1), nitrate reductases (NIR1 and 25299), and a nitrite reductase (NIR2))²⁰ (**Fig R8**).

CRITIQUE 13: *Is there*

a possibility that the LC-UVR treatment in epoch III would have recovered after day 17 of the experiment? Two of the HC-UVR cultures recovered only around day 15 while the third did not recover, similar to the LC-UVR treatment. What if the cells just have needed a couple more days to start growing – this is related to an earlier remark of acclimation strategies of diatoms to UVR.

RESPONSE: We decided to terminate the experiment when there was no recovery in LC cultures after 9 days, whereas the HC cultures had recovered within 5-6 days. Our primary conclusion that *T. pseudonana* has greater resilience in HC conditions would not change, even if the cells in LC conditions needed longer than 9 days for recovery. Our decision to stop the experiment was based on our primary goal to quantify resilience and uncover the systems and molecular underpinnings for increased resilience in HC conditions. Moreover, the design of our study is not ideal to study acclimation to UVR;

the experiment design used in the Sobrino *et al.* study from 2008 is better suited for investigating acclimation strategies to UVR.

CRITIQUE 14: *Since the authors base their assumptions of higher resilience under HC onto a potential down-regulation of the CCM, more information on regulation of certain CCM genes – not only a regulation of CA – should be shown. For example Diatoms have a multitude of CA, HCO₃⁻ and CO₂ transport systems, RubisCO regulation As shown by e.g. Rokitta *et al* 2014 in BMC Genomics (for the cocco. *E.hux*) one can nicely get more detailed understanding from the transcriptome regarding its C acquisition.*

RESPONSE: While we have demonstrated at least 13 genes associated with CCMs were downregulated in response to increased CO₂ levels (see **Supplementary Table S3**)¹², our conclusions also build on reports from other studies. For instance, Kustka *et al.* have conducted a very nice study on the effect of CO₂ on carbon assimilation in *T.pseudonana*²⁵. In that study, Kustka *et al.* used integrated measurements of photosynthetic parameters, transcript abundances and quantitative proteomics to study the effect of a shift to low CO₂ on multiple carbon assimilation mechanisms. Their findings support our claims that lower CO₂ levels do associate with increased transcript or protein abundance of multiple carbon assimilation mechanisms (see Table 2 and Figure 6) in Kustka *et al.* 2014. We have mentioned some of this information in our discussion on page 10, lines 213 and 215-217. Also, see response to reviewer 3's critique 3.

G) **CRITIQUE 15:** *Additional references regaling UVR regulation are needed.*

RESPONSE: Done. See response to critique 7 and page 5, lines 104-112 of the manuscript.

H) *The ms is written clearly yet is missing some aspects as mentioned above.*

Reviewer #3: *(Remarks to the Author):*

*The manuscript by Valenzuela *et al.* describes the response of the diatom *Thalassiosira pseudonana* to UV exposure in cultures cultivated at low CO₂ (300 PPM) and high CO₂ (1000 PPM) conditions. Changes in growth rate and transcriptome shifts are quantified, with the goal of determining how resilient diatoms will be in a future acidified ocean. There has been considerable interest in predicting how ocean organisms will respond to an acidified ocean, with largely detrimental impacts reported for calcifying organisms such as pteropods, corals, and coccolithophores. However not all studies report such detrimental impacts, and this variability makes predicting the biological response to ocean acidification (OA) a continued challenge. Some of the confusion surrounding conflicting reports on the marine biological responses to OA has been generated by technical issues in regulating carbonate chemistry, yet a clear and mechanistic understanding of how different functional types of organisms will respond has not yet emerged. Considering this, the study reported in this manuscript addresses a highly relevant issue.*

How diatoms will respond to an acidified ocean is an outstanding question in the field. It is recognized that the growth rate of diatoms is limited by the supply of CO₂ (Riebesell et al. 1993 Nature 361: 249-251). However this is a complex issue and contradictory results have been published as well (reviewed in Li and Capbell 2013, PLoS ONE). Consequently, it is unsurprising that diatoms may benefit from a future scenario in which OA increases the pCO₂ supply. This study is novel in its attempt to quantify “resilience” of diatoms cultured under low and high CO₂, thereby providing some mechanistic insight into how diatoms may benefit in a proposed future ocean scenario.

Major points:

CRITIQUE 1: *However, this study is not successful in achieving this objective, with the most fundamental issue being a flawed experimental design that does not actually test the effect of UV on *T. pseudonana* cultured at different CO₂ conditions, as it sets out to do.*

RESPONSE: The reviewer might have been disappointed because they expected the manuscript to describe the effect of UVR on *T. pseudonana*, when that was not the main objective of this manuscript. Our primary objective was to develop **a framework to quantify change in resilience** of *T. pseudonana* in response to a projected ecologically relevant increase in CO₂ levels, AND conduct temporal assessment of system-wide changes in gene expression to understand underlying mechanisms. In order to do this, we subjected diatoms to conditions that simulated complex, naturally occurring environmental **changes in multiple factors such as day/night cycles, N availability, and UVR exposure**. The thinking here was that a diatom like any other biological system has to multitask in order to deal with numerous environmental challenges simultaneously. The multitasking **requires *T. pseudonana* to manage trade-offs in allocating resources to different ecologically relevant tasks**, such as shifting between daytime and nighttime physiology, and transitioning between N-replete and deplete conditions. We demonstrated that in absence of UVR stress the diatoms indefinitely sustain growth in this dynamically changing environment (**Fig. 1c and d**). However, the diatom cultures progress towards collapse in a very characteristic and highly reproducible timeframe, when they are required to do all of this multi-tasking while having to respond and recover from a small amount of UVR stress experienced just for an hour each day at noon time. This was a very complex experiment and one that required enormous optimization and attention to detail (a point appreciated by reviewer 2).

This experiment design allowed us to quantify resilience in terms of the number of day/night cycles, and stages over which *T. pseudonana* can recover from UVR stress to sustain growth. **Elevated CO₂ availability shifted this equilibrium and improved resilience of the diatom by enhancing its ability to sustain this taxing growth schedule over a longer number of days and stages (~18 day/night cycles and three stages in HC conditions compared to ~9 day/night cycles and two stages in LC**

conditions). The temporal assessment of transcriptome changes uncovered how the internal transcriptome state of the diatom became uncorrelated with changes in day/night cycles and N availability. Previous studies would not have uncovered this insight because they did not take into account such complex environmental changes; and that is because they utilized a chemostat to grow cultures under constant photoautotrophic conditions^{3,12}.

Thus, we believe this is not a fair assessment and, it appears this might be because the reviewer mistakenly assumed that we set out to test the effect of UVR on *T. pseudonana*.

CRITIQUE 2: *The assertive and colorful title overstates conclusions, which are quickly undermined by the data presented.*

RESPONSE: Our conclusions regarding increased resilience in *T. pseudonana* are supported by our data and analysis, but we submit that we might have over-generalized our findings to all diatoms. We have reworded the title to make it more specific to *T. pseudonana*.

CRITIQUE 3: *The hypothesis that Valenzuela et al. put forth is that energetic savings from reduced CCM expenditures in diatoms at high CO₂ will improve resilience. The literature they cite discusses the CCM in an evolutionarily distinct diatom *Phaeodactylum tricornutum*, and it is a stretch to make similar assumptions about how the CCM operates in the diatom studied here, *T. pseudonana*. In fact the energetic cost model for the biophysical CCM function cited is based on the occurrence of a pyrenoid carbonic anhydrase which is absent in *T. pseudonana* (Tachibana et al. 2011 Photosynthesis Res). Although *T. pseudonana* does have CAs located in the stroma this is an important distinction. It is unclear if the CCM that *T. pseudonana* possesses even requires energy or if it is a passive system. Therefore, this is an unreasonable leap.*

RESPONSE: The reviewer is correct that *T. pseudonana* does not possess a pyrenoid carbonic anhydrase (CA), but not accurate in stating that we base our hypothesis regarding improved resilience on literature that is exclusive to *P. tricornutum*. One of several papers we cite compiled data from four species of diatoms including three centrics from the *Thalassiosira* genus—*T. weissflogii*, *T. pseudonana*, and *T. oceanica*, and one pennate—*Phaeodactylum tricornutum*²⁶. But, again, the reviewer is correct in pointing out that the CCM mechanism in *T. pseudonana* is distinct from the one in *P. tricornutum*. While the latter has a pyrenoid carbonic anhydrase, *T. pseudonana* might make use a modified C4 CCM²⁷, which utilizes a PEP-carboxylase to make oxaloacetate (C4), which is thereafter decarboxylated in the stroma by a pyruvate carboxylase²⁵ (**Fig R9**).

Furthermore, we also base our hypothesis on our recently published work in *Nature Climate Change* in which we reported that elevated CO₂ results in downregulation of a CCM sub-cluster

contains genes encoding plastid-localized membrane proteins, including bestrophin-like proteins, which are homologous to a family of gated anion-selective channels²⁸ permeable to bicarbonate²⁹. These plastid-targeted proteins provide a feasible mechanism for bicarbonate transport to the stroma. Once inside the plastid, bicarbonate can be converted to CO₂ by the stroma-localized³⁰ and constitutively expressed α -CA1 (**Fig. R10**). Interestingly, this carbonic anhydrase is not downregulated by CO₂, suggesting that, unlike *P. tricornutum*³¹ expression levels of a stromal-localized CA are not a point of CCM control in *T. pseudonana*. Another essential feature of efficient CCMs is to prevent the passive diffusion of enhanced CO₂ concentrations out of the cell in general, or the plastid in particular²⁶. The δ -CA3, localized to the plastid membrane space³⁰ and the ζ -CA1, localized to the outer membrane space³⁰ are well placed to serve

this role by converting CO₂ into bicarbonate in the periplastid and periplasmic compartments, where it could be selectively transported back into the cytoplasm or stroma. That these genes are rapidly and sustainably downregulated under elevated CO₂ suggests that preventing diffusive loss of CO₂ is tightly regulated as a point of control for the *T. pseudonana* CCM, in strong agreement with physiology-based model predictions for an efficient diatom CCM²⁶ (**Fig R9**).

regulatory relationships. Abbreviations: SPT/AGT, serine-pyruvate/aspartate aminotransferase; TF, transcription factor; SHMT, serine-glycine hydroxymethyltransferase; GDC(P/T), glycine decarboxylase (P/T) protein; GOX, glycolate oxidase; MFS, major facilitator superfamily; Best, bestrophin superfamily transporter; PGP, phosphoglycolate phosphatase; SLC4, solute carrier family 4 bicarbonate transporter; 3-PGA, 3-phosphoglyceric acid; RuBP, ribulose-1,5-bisphosphate

CRITIQUE 4: *The primary issue with this study is the experimental design and controls. Growth curves don't replicate between the UV challenged and control experiments at both LC and HC conditions during Epoch I, which raises questions and concerns. According to the way the experimental design and results are reported, in Epoch I the pairs of curves plotted in Fig. 1a and 1c in and 1b and 1d should represent the same conditions. However the maximum cell densities in the controls (c and d) appear 2x higher (reached on shorter timescales) than the UV experiments (a and b) (yet in Epoch I where this significant difference arises there has not yet been any UV exposure) ?*

RESPONSE: We have addressed the issue of variability across replicate cultures in our detailed response to reviewer 1's critique 2. In brief, the differences between UVR-treated and control experiments in Fig 1 were because of batch effects, primarily having to do with N-levels. Compiled results from all replicate experiments conducted at different times shows that on average the starting cell densities were comparable across all conditions (**Fig R2**). Please also see our response to critique 6 below, which includes results of a new experiment that was conducted specifically in response to reviewer 3's request. The results show clearly that the collapse phenomenon appears even when cultures achieve high cell density in stage 1 (i.e., without UVR treatment).

CRITIQUE 5: *Further, what is limiting the growth of LC cultures at the end of Epoch I in the UV (which has not yet been treated with UV) and the non UV? If it's N-limitation, as the authors state, then why are neither the control experiment nor the HC cultures similarly N-limited?*

RESPONSE: The cultures are limited for N. Complete consumption of nitrate to undetectable levels (**Fig R3**) and up regulation of N-uptake genes (**Fig R8**) demonstrate that all cultures were N-limited, including control and HC cultures. We have included this data in the manuscript as supplementary figure S1.

CRITIQUE 6: *Most problematic is that authors attribute the impaired growth observed in Epoch II in LC relative to HC to the UV exposure, whereas the simpler (and more likely) explanation is that less healthy cells were transferred between epochs from LC than those in HC. What is likely observed in epoch II is an increase in the length of lag phase in LC which cannot be assigned to the UV exposure treatments. The proper control to test the UV effect would have transferred inocula from the end of each epoch into UV-exposed and non UV-exposed batches, and observed the difference caused by UV alone independent of other variables which influence growth characteristics. This is a major issue with the experimental design that prevents decoupling the effect of LC/HC from UV and therefore the conclusions drawn in the manuscript cannot be adequately evaluated. For example in Epoch II under LC it is critical that we see the impact of the same inoculum exposed to UV and not exposed to*

UV. The Epoch II control used for comparison to the Epoch II UV treatments is problematic because it is not the same inoculum and because the Epoch I cultures in the control and UV (although not yet exposed to UV) are clearly very different.

RESPONSE: In order to attribute impaired growth to UVR, we have performed the experiment requested by reviewer 3. Specifically, at the end of stage 1 (**Fig R11a.i**), culture aliquots were inoculated into fresh media in two different sets of reactors –the first set received standard UVR treatment of 0.5 mW/cm² for one hour at midday (**Fig R11b.i**); and the second set did not receive any UVR treatment (**Fig R11b.ii**). In stage 2, reduced growth rates were observed only in the reactor set that received UVR treatment, with one culture collapsing (**Fig R11b.i**). By contrast, there was no change in growth rates of cultures in the second reactor set that did not receive UVR. In stage 3, we again split inoculum from the UVR-treated stage 2 cultures into two sets of reactors. As expected, all UVR treated cultures in stage 3 (**Fig R11c.i**) failed to re-initiate growth, and untreated cultures retained growth characteristics of the preceding stage 2 cultures (**Fig R11c.ii**). Collectively, these results demonstrate that the collapse phenomenon can be directly attributed to the one hour UVR treatment received by cultures each day at midday. We have included additional text in the manuscript to reflect these findings (see page 7, lines 150-153). Data and plots from the new experiment have been added to the supplement (**Supplementary Figure S2**).

Figure R11. Isolating the role of UVR in collapse of *T. pseudonana* cultures. In **stage 1**, triplicate cultures of *T. pseudonana* were grown under LC (300 ppm CO₂), N-limitation, and 12:12 h light:dark cycles without any UVR exposure (**a.i**). In **stage 2**, aliquots from stage 1 cultures were used to inoculate two sets of triplicate reactors --one set received 0.5mW/cm² UVR for 1-hour each day at noon time (**b.i.**); the second set experienced the same set of conditions as stage 1 cultures (i.e., no UVR) (**b.ii.**). In **stage 3**, aliquots from UVR-irradiated stage 2 cultures were inoculated into two sets of reactors --one set received UVR (**c.i**) and the other did not (**c.ii**). Averaged maximum growth rates of replicate cultures from all three stages with arrows indicating reactor source and destination of each inoculum (**d**). Error bars represent the standard deviation of replicate cell counts (**a-c**) and growth rates (**d**).

CRITIQUE 7: In addition to major differences in yield these cultures appear to have been sampled at different phases of growth (the UV treatment cultures (Epoch I) were clearly sampled in stationary phase but this doesn't appear to be the case the control series). Obviously there are differences between LC and HC in the UV that do not appear as apparent in the controls but it is difficult to decouple the effect of LC and HC from UV without exposure of each LC and HC Epoch inoculum to UV and non UV conditions.

RESPONSE: While the cultures might not have been precisely at the same cell density when they were sampled, we applied constraints that ensured that they were at

comparable phases of growth vis-à-vis N-replete (early) and depleted (late) conditions. Specifically, cultures were sampled as they initiated growth after a lag phase and transferred when nitrate was depleted (less than 1 μM) and Fv/Fm had decreased to a value at or around 0.30. Comparative analysis of transcriptomes from different growth phases, day/nighttime states, across replicate cultures and stages of the experiment demonstrate striking similarity of transcriptomes from comparable conditions (**Fig R7**, also see response to **reviewer 2, critique 11**).

CRITIQUE 8: *Because the experimental design is plagued with aforementioned issues, the interpretation of state shifts obtained by transcriptome sequencing is compromised & difficult to interpret.*

RESPONSE: We have demonstrated that the experiment design had no issues, that the collapse phenomenon can be directly associated with UVR treatment, and that results were highly reproducible (see response to critiques 1 through 7 above).

CRITIQUE 9: *It does seem apparent and convincing that there is a quantifiable loss of “relational resilience” observed in LC cultures, however as previously discussed, the UV stress effect cannot be decoupled from normal batch culture resource limitation dynamics. It’s unclear why quantifying state transitions and relational resilience is better than other more well accepted metrics for fitness or survivability such as cell viability (i.e. with SYTOX Green) or LD50. It would however been informative to establish linkages between such physiological assays and the transcriptomics data.*

RESPONSE: Our responses to reviewer 3’s earlier critiques addresses their concern about attributing loss of resilience to increased stress from repeated exposures to a sub-inhibitory dose of UVR (specifically, see response to **critique 6**).

Our detailed response to reviewer 1 critique 4 and reviewer 3 critique 1 addresses the second question although it warrants some more explanation. Cell survival and viability are properties that assay the degree to which an organism can withstand extreme amounts of a given type of stress. Typically (almost always) an organism can withstand much greater stress than it might encounter in the natural environment. However, knowledge of how much an organism is able to withstand a particular kind of stress does not give insight into its capability to reliably withstand complex multifactorial changes in its highly competitive environment. The efficiency with which it is able to respond and recover from stress will determine whether it will be successful in such a complex environment. That property is described as resilience –it is a well-established concept in the study of complex systems. While viability assays and LD50 help to quantify the amount of a particular kind of stress that would kill an organism, it does not quantify its resilience, and as a result these metrics are unable to predict whether a system is healthy or on the verge of collapse.

CRITIQUE 10: *Finally, this manuscript doesn’t cite appropriate literature. For example, Shi et al. 2010 Effect of Ocean Acidification on Iron Availability in Marine Phytoplankton (Science 327: 676-679) is one of the most highly cited and relevant papers on the subject, but is not referenced or discussed in this manuscript. The synergistic*

environmental changes diatoms will experience in a future acidified ocean should be considered, particularly in a study which claims to predict how these organisms will respond. The idea that diatoms are predicted to face difficulties regarding Fe availability in response to OA conflicts with the claim presented here that diatoms will be more resilient under OA conditions. Studies evaluating the impact of OA on diatoms need to consider this (or at least mention it).

RESPONSE: The reviewer makes an excellent point that we were aware of but did not appropriately address in the manuscript. Limited Fe availability in certain areas of the ocean could increase stress and inhibit growth of *T. pseudonana*. It would be fascinating to investigate how these complex effects of Fe limitation alter diatom resilience. One hypothesis could be that Fe limitation has a net neutral effect on ability of *T. pseudonana* to respond and recover from episodic stress events such as UVR. This is because the effort required to deal with constant Fe stress could consume significant energy savings resulting from relaxed requirement of CCMs. In order to understand how this would play out, i.e., to investigate how such trade-offs in dealing with multiple stressors impinges on organismal resilience, one would need to characterize response-recovery dynamics of *T. pseudonana* in a multi-factorial experiment with combinatorial perturbations in Fe availability and UVR stress. The stress test framework we have developed in this study was designed precisely for this type of an experiment and it was our intention to promote this systems thinking and a systems approach to the study of an emergent property such as resilience. We apologize for our oversight and have now included these discussion points on effect of Fe limitation on resilience with appropriate references on page 3, line 57-58 and page 10, line 224-225.

CRITIQUE 11: *This manuscript also neglects to mention seminal studies from the coccolithophore literature, even in order to contrast the response of diatoms to other phytoplankton which have been more well studied in an OA context. This is an important discussion point to put their results in a broader context.*

RESPONSE: Again, in our zeal to report our exciting results in a concise and easy to understand manner, we were not as comprehensive in discussing the vast literature. Since the reviewer finds that it is important to be more inclusive we have included additional references to put results of our study into a broader context (see page 2 and 3, lines 53-57).

Minor points:

CRITIQUE 12: *Terminology applied to the batch culture experiments is grandiose. Calling growth cycles “epochs” is misleading. It connotes evolutionary timescales and conjures up the sense that within each growth cycle there is genetic adaptation. Can they be termed cycles?*

RESPONSE: We have changed “epoch” to “stage”, which is the terminology used for other stress tests, such as the Bruce protocol for cardiac stress test^{32,33}.

CRITIQUE 13: *Similarly, calling inoculum transfer a “bottleneck” (line 104-105) also implies some significance evolutionarily which is unnecessary and obfuscatory. Or this*

needs to be much better justified or explained is there is some specific point being made.

RESPONSE: We have changed “bottleneck” to “dilution” (see page 6, lines 128 and 130).

CRITIQUE 14: *A description of curve fitting to growth data is not provided and in some cases it looks like S-curves were forced to fit which misleads the reader into thinking that stationary phase was reached. There are also no error bars on cell counts (hemocytometer counts are notoriously variable and models aimed at fitting growth curves should account for this).*

RESPONSE: A short description was provided originally in the Methods section along with a link to the analysis source code. The data were fit using a standard logistic growth curve model³⁴ (added as supplementary eq.1) which utilizes the parameters of growth rate, carrying capacity, and lag phase. Additional detail has been added to the Methods section to help clarify model inputs (page 11 and 12, lines 254-257). All hemocytometer cell counts values reported are averages of four 1 mm² square grids. We did not see much variability between cell counts. We considered biological replication as a more important measure and a rigorous assessment of variability. However, we made sure to calculate the standard deviation for each cell count per replicate and applied the growth model to the additional experiment we performed (R11).

CRITIQUE 15: *Authors state that N is depleted below the detection limit (lines 103-104) and should include these data.*

RESPONSE: Done –See reviewer 2, critique 12 and new supplementary figure S1.

RESPONSE REFERENCES

1. Van Nes, E. H. & Scheffer, M. Slow recovery from perturbations as a generic indicator of a nearby catastrophic shift. *The American Naturalist* (2007).
2. Dai, L., Vorselen, D., Korolev, K. S. & Gore, J. Generic indicators for loss of resilience before a tipping point leading to population collapse. *Science* **336**, 1175–1177 (2012).
3. Veraart, A. J. *et al.* Recovery rates reflect distance to a tipping point in a living system. *Nature* **481**, 357–359 (2011).
4. Song, H. S., Renslow, R. S., Fredrickson, J. K. & Lindemann, S. R. Integrating Ecological and Engineering Concepts of Resilience in Microbial Communities. *Frontiers in Microbiology* **6**, 304 (2015).
5. Shade, A., Peter, H., Allison, S. D., Baho, D. L. & Berga, M. Fundamentals of microbial community resistance and resilience. *frontiers in microbiology* **3**, 1–19 (2012).
6. Gunderson, L. H. Ecological resilience--in theory and application. *Annual review of ecology and systematics* **31**, 425–439 (2000).
7. Botton, S., van Heusden, M., Parsons, J. R., Smidt, H. & van Straalen, N. Resilience of microbial systems towards disturbances. *Critical Reviews in Microbiology* **32**, 101–112 (2008).
8. Ha, S. Y. *et al.* Photoprotective function of mycosporine-like amino acids in a

- bipolar diatom (*Porosira glacialis*): evidence from ultraviolet radiation and stable isotope probing. *Diatom Research* **29**, 399–409 (2014).
9. Liu, Q., Nishibori, N., Imai, I. & Hollibaugh, J. T. Response of polyamine pools in marine phytoplankton to nutrient limitation and variation in temperature and salinity. *Mar. Ecol. Prog. Ser.* **544**, 93–105 (2016).
 10. Lomas, M. W. & Glibert, P. M. Comparisons of nitrate uptake, storage, and reduction in marine diatoms and flagellates. *Journal of Phycology* (2000).
 11. Dai, L., Korolev, K. S. & Gore, J. Slower recovery in space before collapse of connected populations. *Nature* **496**, 355–358 (2013).
 12. Hennon, G. M. M., Quay, P., Morales, R. L., Swanson, L. M. & Virginia Armbrust, E. Acclimation conditions modify physiological response of the diatom *Thalassiosira pseudonana* to elevated CO₂ concentrations in a nitrate-limited chemostat. *Journal of Phycology* **50**, 243–253 (2014).
 13. Holling, C. S. Resilience and stability of ecological systems. *Annual review of ecology and systematics* (1973).
 14. Fuhrman, J. A., Cram, J. A. & Needham, D. M. Marine microbial community dynamics and their ecological interpretation. *Nature Publishing Group* **13**, 133–146 (2015).
 15. Coesel, S. *et al.* Diatom PtCPF1 is a new cryptochrome/photolyase family member with DNA repair and transcription regulation activity. *EMBO reports* **10**, 655–661 (2009).
 16. Rijstenbil, J. W. Assessment of oxidative stress in the planktonic diatom *Thalassiosira pseudonana* in response to UVA and UVB radiation. *Journal of Plankton Research* (2002).
 17. Hennon, G. M. M. *et al.* Diatom acclimation to elevated CO₂ via cAMP signalling and coordinated gene expression. *Nature Climate Change* **5**, 761–765 (2015).
 18. Trimborn, S. *et al.* Inorganic carbon acquisition in potentially toxic and non-toxic diatoms: the effect of pH-induced changes in seawater carbonate chemistry. *Physiol Plant* **133**, 92–105 (2008).
 19. Fielding, A. S., Turpin, D. H. & Guy, R. D. Influence of the carbon concentrating mechanism on carbon stable isotope discrimination by the marine diatom *Thalassiosira pseudonana*. *Can. J. Bot.* (1998).
 20. Ashworth, J., Turkarslan, S., Harris, M., Orellana, M. V. & Baliga, N. S. Pan-transcriptomic analysis identifies coordinated and orthologous functional modules in the diatoms *Thalassiosira pseudonana* and *Phaeodactylum tricorutum*. *Marine Genomics* 1–8 (2015).
 21. Wu, Y., Campbell, D. A. & Gao, K. Faster recovery of a diatom from UV damage under ocean acidification. *Journal of Photochemistry and Photobiology B: Biology* **140**, 249–254 (2014).
 22. Turkarslan, S. *et al.* Mechanism for microbial population collapse in a fluctuating resource environment. *Molecular Systems Biology* **13**, 919 (2017).
 23. Gao, K. *et al.* Rising CO₂ and increased light exposure synergistically reduce marine primary productivity. *Nature Climate Change* **2**, 1–5 (2012).
 24. Drake, J. M. & Griffen, B. D. Early warning signals of extinction in deteriorating environments. *Nature* **467**, 456–459 (2010).
 25. Kustka, A. B. *et al.* Low CO₂ results in a rearrangement of carbon metabolism to support C4 photosynthetic carbon assimilation in *Thalassiosira pseudonana*. *New Phytologist* **204**, 507–520 (2014).
 26. Hopkinson, B. M., Dupont, C. L., Allen, A. E. & Morel, F. M. M. Efficiency of the CO₂-concentrating mechanism of diatoms. *Proc. Natl. Acad. Sci. U.S.A.* **108**, 3830–3837 (2011).

27. Hopkinson, B. M., Dupont, C. L. & Matsuda, Y. The physiology and genetics of CO₂ concentrating mechanisms in model diatoms. *Current Opinion in Plant Biology* **31**, 51–57 (2016).
28. Kane Dickson, V., Pedi, L. & Long, S. B. Structure and insights into the function of a Ca²⁺-activated Cl⁻ channel. *Nature* **516**, 213–218 (2014).
29. Qu, Z. & Hartzell, H. C. Bestrophin Cl⁻ channels are highly permeable to HCO₃⁻. *AJP: Cell Physiology* **294**, C1371–C1377 (2008).
30. Samukawa, M., Shen, C., Hopkinson, B. M. & Matsuda, Y. Localization of putative carbonic anhydrases in the marine diatom, *Thalassiosira pseudonana*. *Photosynthesis Research* **121**, 235–249 (2014).
31. Ohno, N. *et al.* CO₂-cAMP-responsive cis-elements targeted by a transcription factor with CREB/ATF-like basic zipper domain in the marine diatom *Phaeodactylum tricornutum*. *Plant Physiology* **158**, 499–513 (2012).
32. Bruce, R. A., Lovejoy, F. W., Jr & Pearson, R. Normal respiratory and circulatory pathways of adaptation in exercise. *Journal of Clinical Investigation* (1949).
33. Bruce, R. A., Pearson, R. & Lovejoy, F. W., Jr. Variability of respiratory and circulatory performance during standardized exercise. *Journal of Clinical Investigation* (1949).
34. Kahm, M. & Hasenbrink, G. grofit: Fitting Biological Growth Curves with R. *Journal of Statistical Software* **33**, (2010).

Reviewers' comments:

Reviewer #1 (Remarks to the Author):

The authors did a thorough revision and rebuttal, including additional experiments. Many of my initial comments have been adequately addressed (UV levels, resilience terminology, novelty and minor comments), but I still have a major concern regarding the growth curves obtained in the resilience experiment.

Despite the additional experiment as suggested by R3, I am still concerned about reproducibility and validity of the resilience experiment, due to large variation between control and pre-stress treatment samples. Figure 1 shows growth curves of control and treatment experiments, with large variation between controls and treatments, even before initiation of UV stress.

Although in their rebuttal the authors indeed show batch effects can cause variation in carrying capacity, this in my opinion does not warrant the use of control batches having twice the cell density of the treatment at the beginning of the experiment, making me doubt the health of the LC culture, and thus conclusions about their resilience (Lines 133-142). They aimed to address this in figure R11, which shows there was a UV effect, but in relation to resilience differences with HC it would have made sense to show these data for HC as well.

Additionally, plotting controls and treatments using different x and y-axes in Fig. 1 is misleading, and it seems that controls were run faster than treatment cultures.

Transcriptomic differences are interesting, and can give an indication of population health, but perhaps using them as an early warning signal is not very feasible, as one would prefer something fairly straightforward to measure.

Reviewer #2 (Remarks to the Author):

The authors did a good job revising the MS according to most of the reviewer's remarks. I find the dataset presented in this manuscript to be very comprehensive, yet highly complex and not easy to understand for a general readership. If there is any way to rewrite parts of the MS for the general public/non specialists – I (and they) would appreciate it.

General remarks:

The authors did a good job describing OA responses as well as nutrient decline under future scenarios in the introduction. The authors should, however, try and add language describing how this laboratory study is linked to the field – giving a “real-life” applicability of the experimental design (e.g. describing the conditions under which the cells might see OA, NO₃, and especially UV stress in the field and/or referencing the conditions to field studies (e.g. adding information to the early parts of the MS from line 155 of the MS)).

Additional remarks:

Line 106: what do the authors mean with 6,4 photoproducts?

Fig: 2:

I do not understand the way max. growth rate is shown in this Figure as well as in Figure S3b. What is the unit trying to indicate? Growth is given as μ or doubling time and not as an increase in cell number per volume per day. Please revise.

There seem to be large differences in growth rate when comparing Fig 2 and Fig S3. Please explain.

Fig. 3 and 4. I can not comment on those figures and have to rely on my reviewer colleagues to comment on these.

Fig S1. As requested, the authors now show NO₃ drawdown. HOWEVER – the show this as a percent. I assume they used % of the initial from each growth curve. This means – in my view – nothing. The concentration is important here – For example – if the NO₃ concentration was Only 50% in Stage 2 of Figure S1a compared to stage 1 – this can explain the reduced carrying capacity. Since carrying capacity is important to understand the take home message of the whole paper I do request the concentration of NO₃ shown in these figures rather than the % of concentration.

Fig. S2: I'm still puzzled by the different responses of the triplicate cultures (S2 b-i). Is this a culture artefact/bottle effect? Please explain.

Reviewer #3 (Remarks to the Author):

The revised manuscript addresses many of the concerns raised in the initial review and the authors have adequately addressed the primary concern re: experimental design with additional experimentation and inclusion of these data in the manuscript. However, the manuscript still relies too heavily on the assumption that the loss of relational resilience is solely attributable to the added energetic burden imposed by a CCM at low CO₂ (initial critique 3).

As the authors acknowledge, the CCM in *T. pseudonana* is not fully characterized making it difficult to predict what the energetic cost truly is (and calling into question the relevance of assumptions of energy cost in *T. pseudonana* based on Hopkinson et al. 2011). While it is reasonable to assume that the energetic burden of inducing a CCM may be one of the reasons that resilience is compromised at low CO₂, are there not other feasible explanations (ex: increased metabolic cost of processing glycolate from photorespiration)? Along these lines, it is suggested that the text is revised in specific places to reduce attributing the energetic savings to the CCM. For example, suggest replacing text such as "either directly or indirectly from downregulation of CCMs under elevated CO₂" (lines 98-99) with text like "either directly or indirectly from energetic savings associated with growth at elevated CO₂ [one possible factor being a downregulation of the CCM]".

Speaking to the specific points raised about the *T. pseudonana* CCM in the author's rebuttal - the papers cited (Hennon et al. 2015, Kustka et al. 2014) only hypothesize the existence of alternate CCMs (bestrophin channels, C4-type biochemical CCM). The bestrophin genes are not functionally characterized in *T. pseudonana*, and though the study cited (Qu & Hartzell 2008) has found these channels to be permeable to HCO₃, these genes are mostly thought to function as Cl⁻ ion channels in other systems. The substrate specificity/functionality of the *T. pseudonana* homologs is not known, and the authors should be cautioned against making conclusions about the functionality of these genes based solely on co-expression with other CO₂-sensitive genes and an undemonstrated degree of homology to more well-characterized bestrophin genes. Also, there are issues with the C4 CCM proposed by Kustka et al. 2014, since it requires specific conditions to drive the pyruvate carboxylase in the reverse direction of what is thermodynamically favorable. In that scheme, ATP is produced both from the pyruvate kinase reaction as well as in the reverse pyruvate carboxylase reaction – potentially making this CCM an energetically productive cycle (seemingly undermining the authors' assumption that the CCM is costly). Ultimately, the primary issue with how this topic is presented in the

manuscript currently is that it is misleading to the reader to base conclusions on an incomplete understanding of how the *T. pseudonana* CCM works, without more lengthy discussion of these nuances in the text.

This is an important point, because the role of the CCM and the cost of concentrating carbon in this organism is referenced several times throughout the manuscript (abstract: p2 lines 45-47, p4 lines 92-99, p10 lines 212-218). If the authors believe strongly that the downregulation of the CCM is the sole/primary source of energetic release at high CO₂, the case must be made more convincingly. However, emphasizing the CCM may not be important to the overall conclusions drawn, and this concern may be addressed simply by improving the precision of the language used.

Minor points:

Line 111: the wording "dissect the combinatorial effects" suggests that the effect of individual parameters on resilience were assessed which is not accurate. Would be better to change to "assess the combined effects"

Line 175: typo ", ,"

Line 270: Include SRA accession

Summary of response:

The reviewers were very enthusiastic about the manuscript and they were generally pleased with our response to their critiques. However, at least one reviewer felt strongly, and the editor concurred, that some concerns needed additional work and analysis. We have now addressed all of the remaining reviewer concerns through additional experimentation, analysis, and by revising and restructuring the manuscript. We submit that the reviewers' thoughtful comments, suggestions and critiques were well founded and the edits have made the manuscript significantly better. Here is a brief summary of the major revisions:

- 1) **Performed new experiments to address discrepancy in carrying capacity of control cultures relative to stress-test cultures.** The reviewers correctly noted that the control cultures had significantly higher carrying capacities and growth rates compared to stress-test cultures in stage 1. They were concerned if the lower carrying capacity and growth rate might have contributed to the collapse of cultures in the stress test. Upon further investigation, we figured out that the discrepancy in culture characteristics was due to higher nitrate levels in the media batch that was used to grow the controls. We have now repeated two control experiments at 1,000 ppm (high carbon: HC) and 300 ppm (low carbon: LC) CO₂ with the same starting concentration of nitrate (~65 μM) as in the stress test and as expected the carrying capacities, growth rates, and nitrate drawdowns are now comparable—and as before the control cultures never collapsed in either LC or HC condition (**Fig. R1 [Supplementary Figure S1]**).
- 2) **Revised and restructured manuscript to make it more accessible to both generalist and specialist readers.** As requested by reviewers we have restructured and revised the manuscript to make the results more accessible to a general readership. We also made adjustments to our language in describing the effects of CCMs on the resiliency of *T. pseudonana*, as suggested by Reviewer 3. We acknowledge and are in agreement that downregulation of CCMs play a role but are not the sole factor in the increased resilience of the diatom and is a part of the global response to growth at elevated CO₂.
- 3) **Remade figures with standard practices, nomenclatures, units and axes.** As requested by the Reviewer 2, we now report nitrate values as (μM) instead of percentages, and growth rates as specific growth rates [μ (d⁻¹)] which is common practice (**Fig. 2, S2, and S3**).

We have made a sincere effort to address each reviewer comment and are confident that the new experiments, analysis, and revisions will remove any lasting concerns. Below we provide a point-by-point response to each critique.

From the Editor:

COMMENT: *The reviewers find substantial improvement in the revision including, to some extent, the additional experiments. However, it does not appear that the new work has alleviated the critical concerns about the differences in population growth between the initial LC experimental and control groups. These reservations cast doubt on the strength of the novel conclusions that can be drawn at this stage, and are sufficiently important to preclude publication of this study in Nature Communications.*

RESPONSE: In order to definitively show that carrying capacity had no effect on when cultures collapse, we repeated the two (LC and HC) control experiments with initial nitrate concentrations of ~65 μM, as in the stress-test experiments. Carrying capacities and growth rates of control cultures in the new experiments performed over 20 days, and 4 stages, were similar to those of the treatment groups.

LC and HC control cultures maintained consistent growth rates through all stages, with higher carrying capacities under HC conditions, as expected (**Fig R1a and b; Fig. S1 in revised manuscript**). Nitrate draw down profiles through exponential and stationary phases of growth was also similar (Fig R1-dashed black lines) to treatment cultures. Relative to HC control cultures, growth dynamics of LC control cultures became slightly more variable across replicates in later stages, but none of the replicate cultures collapsed. The increased variability in growth dynamics of LC cultures suggested that *T. pseudonana* is more sensitive to fluctuating environments in LC conditions and, therefore, potentially more susceptible to the addition of another ecologically relevant stress, which explains why they succumb to very low doses of UVR in the stress-test. These new results directly address the concern regarding differences in population growth across control and treatment groups, and reinforce the central conclusion that *T. pseudonana* is more resilient under HC conditions.

Fig. R1 [Supplementary Figure S1]. Repeated growth dynamics and nitrate profiles for LC and HC cultures during the non-UVR conditions and stress-test.

Reviewer #1:

The authors did a thorough revision and rebuttal, including additional experiments. Many of my initial comments have been adequately addressed (UV levels, resilience terminology, novelty and minor comments), but I still have a major concern regarding the growth curves obtained in the resilience experiment.

CRITIQUE 1: *Despite the additional experiment as suggested by R3, I am still concerned about reproducibility and validity of the resilience experiment, due to large variation between control and pre-stress treatment samples. Figure 1 shows growth curves of control and treatment experiments, with large variation between controls and treatments, even before initiation of UV stress...*

RESPONSE: See response to editor. **-Done**

CRITIQUE 2: *They aimed to address this in figure R11, which shows there was a UV effect, but in relation to resilience differences with HC it would have made sense to show these data for HC as well.*

RESPONSE: The aim of figure R11 (**Fig. S2 in revised manuscript**) was to determine whether or not UVR alone was the sole cause of the decrease in growth dynamics during stage 2 (0.5 mW/cm²) and not the health of the transferred population, in which case the former was true. We showed that, in fact, UVR alone was the cause for the decrease in carrying capacity and growth rate using LC cultures. Thus, repeating this experiment with HC cells would not alter this conclusion. Additionally, Reviewer 3 who initially called for this specific experiment found it to be sufficient in addressing this concern, “*the authors have adequately addressed the primary concern re: experimental design with additional experimentation and inclusion of these data in the manuscript.*” **-Done**

CRITIQUE 3: *Additionally, plotting controls and treatments using different x and y-axes in Fig. 1 is misleading, and it seems that controls where run faster than treatment cultures.*

RESPONSE: We have incorporated the new repeated control experiments (**Fig. 1a and b; Fig. S1 in revised manuscript**) and have changed both the x and y-axes scales. The repeated control experiments last approximately 20 days each, over 4 stages, for both HC and LC conditions. Each control experiment was performed with similar initial nitrate concentrations as the stress-tested treatment cultures, resulting in similar growth rates, and carrying capacities. **-Done**

CRITIQUE 4: *Transcriptomic differences are interesting, and can give an indication of population health, but perhaps using them as an early warning signal is not very feasible, as one would prefer something fairly straightforward to measure.*

RESPONSE: The reviewer is correct about the interesting nature of the transcriptomic differences but may be misinterpreting our conclusions. The diatom must transition between and restore the four principal physiological states based on their environmental condition (i.e. light, dark, nutrient replete and nutrient deplete). As the cultures approached collapse, the transitions between the physiological states of replicate cultures became significantly more variable and uncorrelated with the environment (i.e., they lost *relational resilience*). The transcriptomic analysis reveals “loss of relational resilience” as the potential physiological mechanism explaining why the diatom collapsed. Early warning signals have been identified in multiple systems and can be characterized as an increase in autocorrelation or variability between a set of indicators¹⁻⁵. We see this phenomenon appear in the stage prior to collapse, i.e. in stage 2 for LC conditions and stage 3 for HC conditions, in both the increased variability of the transcriptomes and photosynthetic efficiencies.

The early warning signals we observed in our experiment may not be ideal for field monitoring as it stands, but that should not discount the value of the finding. The strengths of this manuscript are in the novel experimental framework and analysis tools for predicting resilience and determining state-descriptors. We hope these findings could be the foundation for conducting additional stress-tests and experiments on multiple parameters and microbial communities to find more straightforward indicators that can then be easily deployed in the field. That being said we did reduce the emphasis of “loss of relational resilience” as the early warning signal to diatom collapse in the abstract. **-Done**

Reviewer #2:

CRITIQUE 1: *I find the dataset presented in this manuscript to be very comprehensive, yet highly complex and not easy to understand for a general readership. If there is any way to rewrite parts of the MS for the general public/non specialists – I (and they)would appreciate it.*

RESPONSE: We have made substantial alterations to the manuscript that including rewording and structural rearrangements to address this concern. Based on these changes we are confident that the revised manuscript will be easily understood by a general readership as well as specialists. **-Done**

CRITIQUE 2: *The authors should, however, try and add language describing how this laboratory study is linked to the field – giving a “real-life” applicability of the experimental design (e.g. describing the conditions under which the cells might see OA, NO₃, and especially UV stress in the field and/or referencing the conditions to field studies (e.g adding information to the early parts of the MS from line 155 of the MS))*

RESPONSE: The reviewer makes a reasonable request and we have restructured the manuscript in order

to highlight how this particular laboratory study provides a basis for exploring these phenomena. In our conclusions, we address how this framework can be developed and adapted to quantify the consequences of changes in diverse environmental factors on other organisms or microbial communities. This is the real strength of the manuscript and should provide a foundation for further research in this area. **-Done**

CRITIQUE 3: Line 106: *what do the authors mean with 6,4 photoproducts?*

RESPONSE: We are referring to pyrimidine dimers caused by UVR induced crosslinking of neighboring pyrimidines. We have changed the manuscript to the more general term of cyclobutane pyrimidine dimers (page 6, line 140). **-Done**

CRITIQUE 4: Fig: 2: *I do not understand the way max. growth rate is shown in this Figure as well as in Figure S3b. What is the unit trying to indicate? Growth is given as μ or doubling time and not as a increase in cell number per volume per day. Please revise.*

RESPONSE: We used a logistic curve fitting model to visualize the growth dynamics of the diatoms over day-night cycles. From this model, we calculated the maximum growth rate as a function of cell concentration over time. In the revised manuscript, we report growth rates as specific growth rate (μ). We calculated μ from linear regression of the natural log of cell concentrations versus time (days) over the exponential phase of growth⁶. We have revised Fig. 2, S2, and S3 to reflect this request. **-Done**

Fig. R4 [Supplemental Figure S2]. Isolating the role of UVR in collapse of *T. pseudonana* cultures.

CRITIQUE 5: There seem to be large differences in growth rate when comparing Fig 2 and Fig S3. Please explain.

RESPONSE: The difference in growth rates between Fig. 2 and Fig. S3 had to do with how they were calculated from fitted logistic curves. In response to Reviewer 2's request (critique 4) we have recalculated specific growth rate [μ (d^{-1})], which is similar across all experiments we have performed and comparable to previously reported growth rates for diatoms^{6,7}. **-Done**

CRITIQUE 6: Fig S1. As requested, the authors now show NO_3 drawdown. HOWEVER – the show this as a percent. I assume they used % of the initial from each growth curve. This means – in my view – nothing. The concentration is important here – For example – if the NO_3 concentration was Only 50% in Stage 2 of Figure S1a compared to stage 1 – this can explain the reduced carrying capacity. Since carrying capacity is important to understand the take home message of the whole paper I do request the concentration of NO_3 shown in these figures rather than the % of concentration.

RESPONSE: We now show nitrate concentrations in μM (see Fig R1 above, and Fig. S1 in revised manuscript). **-Done**

CRITIQUE 7: *Fig. S2: I'm still puzzled by the different responses of the triplicate cultures (S2 b-i). Is this a culture artefact/bottle effect? Please explain.*

RESPONSE: No this is not an artefact. This is a classic response of any complex system that is close to a bifurcation point. This phenomenon has been observed and documented across many complex systems; and specifically, it has been shown that dilution of resource-limited populations amplifies variability across replicate cultures, and when pushed to the extreme with progressively greater stress events the cultures approach a bifurcation point and may **collapse at different times depending on subtle differences in their respective histories**^{1,4,8-10}. We highlight this point in the manuscript (page 6, lines 129-135). In the stress-test, diatoms are transferred when the nitrate in the media has been consumed (<1 μM) and the photosynthetic efficiency (F_v/F_m) is approximately 0.3 (i.e. late exponential or stationary phase). We then transfer that population into fresh medium ($\sim 65 \mu\text{M}$ nitrate) and effectively dilute the cells to 10^5 cells/mL, to test their ability to recover and re-establish population growth. While the control cultures can withstand this repetitive dilution and re-growth, going through four different physiological states without collapse (see Fig. 1a and b), however in the stress-test, the addition of a small amount of UVR at midday for 1 hour amplifies subtle differences in the ability of replicate cultures to respond and recover from stress (i.e. S2 b.i). Repeated application of UVR each day, and across multiple stages pushes the cultures to a bifurcation point—although all cultures collapse eventually, they do so at different times because of this amplified variability. This variability manifests in photosynthetic efficiency, lag-time, and also in the overall transcriptome state. Importantly, this is a reproducible phenomenon that we (and others) have observed; in our case, we reproducibly observe that HC cultures are able to withstand conditions of the stress-test over a longer time frame relative to LC cultures, leading to our conclusion that *T. pseudonana* will become significantly more resilient in an acidified ocean.

Reviewer #3:

The revised manuscript addresses many of the concerns raised in the initial review and the authors have adequately addressed the primary concern re: experimental design with additional experimentation and inclusion of these data in the manuscript.

CRITIQUE 1: *However, the manuscript still relies too heavily on the assumption that the loss of relational resilience is solely attributable to the added energetic burden imposed by a CCM at low CO₂ (initial critique 3). As the authors acknowledge, the CCM in *T. pseudonana* is not fully characterized making it difficult to predict what the energetic cost truly is (and calling into question the relevance of assumptions of energy cost in *T. pseudonana* based on Hopkinson et al. 2011). While it is reasonable to assume that the energetic burden of inducing a CCM may be one of the reasons that resilience is compromised at low CO₂, are there not other feasible explanations (ex: increased metabolic cost of processing glycolate from photorespiration)? Along these lines, it is suggested that the text is revised in specific places to reduce attributing the energetic savings to the CCM. For example, suggest replacing text such as “either directly or indirectly from downregulation of CCMs under elevated CO₂” (lines 98-99) with text like “either directly or indirectly from energetic savings associated with growth at elevated CO₂ [one possible factor being a downregulation of the CCM]”.*

RESPONSE: We agree with the reviewer that the downregulation of CCMs is a factor for the greater resilience in HC cells but might not be the sole reason. We have made an earnest effort to attribute the increased resilience to growth at elevated CO₂, while still highlighting the importance of the downregulation of CCMs as being one possible aspect of a global response leading to the difference in *T. pseudonana* sensitivities. Please see pages 2, 4, 10; lines 45-48, 86-88, 223-231— respectively.

CRITIQUE 2: *Ultimately, the primary issue with how this topic is presented in the manuscript currently is that it is misleading to the reader to base conclusions on an incomplete understanding of how the *T. pseudonana* CCM works, without more lengthy discussion of these nuances in the text. This is an important point, because the role of the CCM and the cost of concentrating carbon in this organism is referenced several times throughout the manuscript (abstract: p2 lines 45-47, p4 lines 92-99, p10 lines 212-218). If the authors believe strongly that the downregulation of the CCM is the sole/primary source of energetic release at high CO₂, the case must be made more convincingly. However, emphasizing the CCM may not be important to the overall conclusions drawn, and this concern may be addressed simply by improving the precision of the language used.*

RESPONSE: We agree with the reviewer's points and have revised the manuscript with more precise language per their recommendation. Please see Reviewer 3 critique 1 as well as pages 2, 4, 10; lines 45-48, 86-88, 223-231— respectively.

MINOR POINTS:

CRITIQUE 3: *Line 111: the wording “dissect the combinatorial effects” suggests that the effect of individual parameters on resilience were assessed which is not accurate. Would be better to change to “assess the combined effects”*

RESPONSE: -Done

CRITIQUE 4: *Line 175: typo “, ,”*

RESPONSE: -Done

CRITIQUE 5: *Line 270: Include SRA accession*

RESPONSE: All sequences have been made publicly available through the National Center for Biotechnology (NCBI) Sequence Read Archive (SRA), BioProject PRJNA386016 and are referenced in the manuscript (page 12, lines 278-280). -Done

RESPONSE REFERENCES

1. Veraart, A. J. *et al.* Recovery rates reflect distance to a tipping point in a living system. *Nature* **481**, 357–359 (2011).
2. Drake, J. M. & Griffen, B. D. Early warning signals of extinction in deteriorating environments. *Nature* **467**, 456–459 (2010).
3. Scheffer, M. *et al.* Early-warning signals for critical transitions. *Nature* **461**, 53–59 (2009).
4. Dai, L., Vorselen, D., Korolev, K. S. & Gore, J. Generic indicators for loss of resilience before a tipping point leading to population collapse. *Science* **336**, 1175–1177 (2012).
5. Dakos, V. & Bascompte, J. Critical slowing down as early warning for the onset of collapse in mutualistic communities. *Proc. Natl. Acad. Sci. U.S.A.* **111**, 17546–17551 (2014).
6. Shi, D., Xu, Y., Hopkinson, B. M. & Morel, F. M. M. Effect of Ocean Acidification on Iron Availability to Marine Phytoplankton. *Science* **327**, 676–679 (2010).
7. Gao, K. *et al.* Rising CO₂ and increased light exposure synergistically reduce marine primary productivity. *Nature Climate Change* **2**, 1–5 (2012).
8. Dai, L., Korolev, K. S. & Gore, J. Slower recovery in space before collapse of connected populations. *Nature* **496**, 355–358 (2013).
9. Axelrod, K., Sanchez, A., Gore, J. & Ferrell, J. Phenotypic states become increasingly sensitive to perturbations near a bifurcation in a synthetic gene network. *eLife Sciences* **4**, e07935 (2015).

10. Rindi, L., Bello, M. D., Dai, L., Gore, J. & Benedetti-Cecchi, L. Direct observation of increasing recovery length before collapse of a marine benthic ecosystem. *Nat. ecol. evol.* **1**, 0153 (2017).

Reviewers' comments:

Reviewer #2 (Remarks to the Author):

Re-review of the manuscript by Dr Baliga and co-workers entitled "The marine diatom *Thalassiosira pseudonana* is more resilient in an acidified ocean".

The authors invested significant work to revise this manuscript. They addressed all of my comments well and made significant changes to improve the quality of the manuscript. The additional experiments performed enhanced the quality of the data significantly and the rewriting of the manuscript resulted in a much better reading experience.

This paper is important in it's work as it shows that cells not only can be persistently affected from "genetic change but also from global dysregulation of cellular processes" – as stated in line 221-223. This "stress" factor is important to consider in lab experiments (long term acclimation) but more importantly can affect phytoplankton succession patterns in nature and thus affect food web structures and biogeochemical cycles.

I congratulate the authors to this well performed experiment and hope to see this paper published and presented at conferences.

My only minor comment is based on the first submission of the data. (This might be the wrong forum for this comment but given the influence some of the authors of this manuscript have on the community I wish to address it.) Why did the authors not realize the difference in nutrient concentration in the media beforehand? I understand that culturing phytoplankton comes with challenges and quite some paper are published with data affected by small irregularities in culture media. I ask the PIs and the students involved in this manuscript to check for these avoidable issues in future experiments, e.g. grow a batch culture and check for carrying capacity, measure N, P, Si in the media, regularly check for pH... .

****Additional comments to the authors regarding Reviewer 3's report****

Some suggestions for improving the dataset and maybe address the concerns of Reviewer 3 follow below:

Based on the experiments from the authors as well as known data, I feel that the authors could revise parts of their results to strengthen the study.

- I think the authors can reevaluate data from the transcriptomic results. This could include transcriptomic regulations of the UV treatment which might bear some indication on how the cells become more resilient.
- The authors should mention and implement data on T.p. CCM (e.g. from Clement et al 2016 and 2017 - not mentioned in this MS). Clement et al 2017 characterized the CCM on a transcriptomic level, hence a direct comparison could be possible (although Clement used extreme pCO₂ levels).
- The authors also should answer the following: Which resources would allow the cells to be more resilient (in addition to the CCM regulation).
 - o The authors also should discuss known processes (not related to the CCM (transporters) or FvFm) which are UV affected, potentially causing results as shown in the study.
- Include more information on UV effects and how cells can cope with it. Maybe refer to "Effects of ultraviolet radiation and CO₂ increase on winter phytoplankton assemblages in a temperate coastal lagoon (Domingues et al 2014)
- Additionally, please add information about effects of UV on RubisCO (Vincent&Neale 2000 -which could affect the carbon metabolism)

Reviewer #3 (Remarks to the Author):

It is clear that the authors have made an earnest effort in addressing reviewer concerns through the inclusion of additional experiments aimed at strengthening their conclusion that elevated CO₂ increases relational resilience in *T. pseudonana*. Unfortunately, while the experimental design has been improved sufficiently to meet general standards of rigor, there are remaining issues with data interpretation, the subsequent conclusions drawn, and a lack of necessary data to support those conclusions and interpretations. In the last review, I emphasized that there are alternate interpretations (beyond energetic cost of the CCM) that should be considered. Though the authors did revise the text to acknowledge the existence of alternate interpretations in the current version of the manuscript, their validity has not been taken seriously. This is likely because these alternate interpretations critically undermine the conclusions on which the impact of the entire manuscript is constructed.

As the authors acknowledge, the experiments test the combined effect of CO₂ level, UV exposure, and N-availability. While it appears to be a straightforward design, it is in reality very complicated to interpret due to the number of variables introduced (that are not tested independently). Authors conclude that LC cultures incur additional "energetic costs" (vague) invoking the cost of the CCM or other C assimilation pathways as an explanation (line 226-227) as the cause of reduced resilience. This is asserted strongly in the final sentence of the abstract (lines 45-49). However, there is a more straightforward interpretation of these results. Since cultures had equal amounts of nitrate, and both drew down this nitrate completely (below detectable levels) but grew to different densities, the LC cells should have a lower C:N than HC cells indicating that LC cells are carbon limited. LC cells probably have a reduced ability to store C (as chrysolaminarin) during growth at LC, leading to this reduced C:N. The finding that a reduction in/disruption of optimal cellular C:N would lead to increased susceptibility to (any kind of) stress is hardly a novel concept in algal physiology. Other possible changes to LC cells such as properties of the light harvesting complex (leading to increased NPQ under LC, leading to antenna shifts that could leave cells more susceptible to UV) have been completely ignored in the interpretation/discussion.

The most novel contribution of the manuscript is the use of transcriptome state changes as a proxy for physiological state (line 224). This analysis is interesting and valuable and is well-presented in the current manuscript. However, the study as is relies too heavily on this transcriptome data and the conclusions are neither strong enough, nor sufficiently validated. The overall impact of the study would be improved greatly if authors were able to specify in mechanistic detail what the energy trade-off was between LC and HC cultures rather than simply speculate. To do this most effectively would require further work to quantify relevant physiological/biochemical parameters (TOC, TON, TC, TN, chrysolaminarin content, carbon fixation rates) or at least relevant molecular characteristics (westerns showing increased CCM components). Growth curves are not sufficient, and the inclusion of biochemical and physiological data is now standard practice in high-impact transcriptome studies.

Overall, the manuscript has been improved by review but does not represent a significant enough advance in our understanding of algal physiology or the algal response to fluctuating environments to merit publication in *Nature Communications*. Largely this results from lack of data relating to an underlying mechanism that would cause the reported observations. Without an adequately evaluated mechanisms there is really not a strong advance.

Minor comment:

Figure 1 and Supplementary Figure 1 are nearly identical, with Supplementary Figure 1 being more informative. Suggest reducing the redundancy of both and replace Figure 1 with Supplementary Figure 1.

Summary of response:

We were very encouraged by Reviewer 2's congratulatory comments, and are greatly appreciative of their thoughtful recommendations on addressing Reviewer 3's remarks. With further guidance from the Editor, we have addressed all of these recommendations in their entirety through additional transcriptomic analysis and making extensive revisions to the manuscript. Here is a brief summary and a few highlights of what we have done:

- 1. Performed additional transcriptome analysis to investigate potential contributions of other genes and processes in relation to increased resilience at HC conditions.** Our previous analysis of transcriptional changes (**Fig. R1**) did not reveal expression patterns that implicated UVR or oxidative stress response genes in increased resilience of diatoms under HC conditions. We have repeated the prior analysis including additional DNA repair genes (**Fig. R2**) and performed rigorous statistical assessment (**Fig. R3**) to investigate whether other genes and processes may have contributed to increased resilience. These analyses have reaffirmed our prior findings and have not changed conclusions of our study. We have added all of these analyses and their outcomes to the Results section.
- 2. Added a citation referencing Clement et al. 2017.** We have compared transcriptome data from our study with protein-level changes reported in Clement et al. 2017 (**Fig. R4**), and discovered that downregulation of carbon acquisition genes under HC conditions is consistent at both transcriptional and translational levels. We have included this citation to the paper.
- 3. Addressed which resources might contribute to increased diatom resilience under HC.** We have significantly expanded discussion on resilience in context of prior work that demonstrate energy savings resulting from downregulation of various processes under elevated CO₂ conditions. Additionally, we have expanded our discussion on collapse of cultures as an emergent phenomenon that results from systems failure due to loss of relational resilience.
- 4. Discussed UV response dynamics in diatoms.** We have expanded the discussion on dynamics and consequences of the UVR response. We also cite the Domingues et al. 2014 study that demonstrates varying effects of environmental factors on different phytoplankton groups (e.g., diatoms, coccolithophores, or cyanobacteria). Importantly, findings from this mesocosm study support our conclusions; specifically, that relative abundance of some diatoms increased under elevated CO₂ and UVR.
- 5. Restructured the manuscript per editorial request.** We have restructured the manuscript into Introduction, Results, Discussion, and Methods sections.

In all, we have expanded discussion on energetics of CCMs, impact of UVR on phytoplankton, and expanded the discussion on why *T. pseudonana* will be more resilient in an acidified ocean.

Reviewer #2:

CRITIQUE 1: *I think the authors can reevaluate data from the transcriptomic results. This could include transcriptomic regulations of the UV treatment which might bear some indication on how the cells become more resilient.*

RESPONSE: During the first round of revisions we performed transcriptome analysis targeting known UVR and oxidative stress response (OSR)-related genes^{1,2} and observed minimal differences across HC and LC conditions at stage 1 and 2 (**Fig. R1**). The initial analysis targeted 46 genes (i.e., peroxidases, glutaredoxins, and superoxide dismutases [SODs]) that are known to respond to reactive oxygen species (ROS) often caused by extreme UVR exposure. Because UVR may also damage DNA directly through the formation of cyclobutane pyrimidine dimers^{3,4}

Figure R1. Differential expression of oxidative stress response genes across stages 1 and 2 in HC and LC conditions. Expression analysis of oxidative stress response genes ($n = 38$) associated with increased production of reactive oxygen species (ROS), including peroxidases, glutaredoxins, and superoxide dismutases. Genes are grouped based on hierarchical clustering, and labeled with their transcript I.D. and putative function. Cells outlined in bold represent genes with significant (p -value < 0.05) differential expression with a \log_2 fold change (HC/LC) set at $+1/-1$. (SODM: superoxide dismutases, SODF: iron superoxide dismutases, PUF: protein of unknown function)

47 we analyzed transcript levels of an additional 64 DNA repair genes, including those involved in
 48 *homologous recombination, base excision repair, mismatch repair, nucleotide excision repairs,*
 49 *and DNA photolyases (Fig. R2).* We did not observe any dramatic changes in transcript levels of
 50 OSR and DNA repair genes across UVR-treated cultures growing in HC and LC conditions.

51 Furthermore, we investigated if the distribution of relative expression changes of UVR-
 52 response genes in HC versus LC conditions was affected by UVR (i.e., stage 2 (0.5 mW/cm²
 53 UVR) with stage 1 (no UVR)). We did not observe a significant difference in distribution of
 54 expression changes of any set of UV-response genes between stages (Fig. R3a and b). Thus,
 55 we conclude that UVR-responsive genes are not responding differentially between carbon
 56 conditions due to UVR exposure during stage 2.

Figure R2. Differential expression of DNA repair pathways across stages 1 and 2 in HC and LC conditions. When UVR damages DNA, repair mechanisms are expressed to limit corruption of the genetic code. Transcriptomic analysis of 69 DNA repair genes (i.e., *homologous recombination*, *base excision repair*, *mismatch repair*, *nucleotide excision repairs*, and *DNA photolyases*) revealed only subtle differences between carbon conditions. Genes are grouped based on hierarchical clustering of their gene expression. Cells outlined in bold represent genes with significant (p -value < 0.05) differential expression with a \log_2 fold change (HC/LC) set at +1/-1.

Figure R3. UVR does not have a significant transcriptional effect on differential carbon response of UVR responsive and central carbon metabolism genes. (a) Distribution of the differential transcriptional response to carbon levels between stage 2 and stage 1 for a set of OSR and DNA damage responding genes ($n = 107$) and central carbon metabolism genes ($n = 102$). Three genes (colored dots) from UVR-response and two in the central carbon metabolism show a differential response, but are not significant (cross-validation; p -value = 0.765 and p -value = 0.8, respectively). (b) Probability distributions of the number of outliers computed from 10,000 random sets of 107 (circles) and 102 genes (squares). Orange and green dashed lines mark the observed outliers for the UVR response and central carbon metabolism gene sets, respectively. At least eight and nine outliers are required to indicate significance in sets of $n = 102$ and $n = 107$, respectively. Grey area indicates cumulative probability ≥ 0.95 .

58 **CRITIQUE 2:** The authors should mention and implement data on *T.p.* CCM (e.g. from Clement
 59 *et al* 2016 and 2017 - not mentioned in this MS). Clement *et al* 2017 characterized the CCM on a
 60 transcriptomic level, hence a direct comparison could be possible (although Clement used
 61 extreme $p\text{CO}_2$ levels).

62 **RESPONSE:** Recent studies by Clement *et al.* have provided new and valuable insights into CCM
 63 function as well as the indirect metabolic responses to low and elevated CO_2 concentrations.
 64 Although their two studies utilize an acute and extreme CO_2 shift from 20,000 ppm to 50 ppm,
 65 under constant light conditions, their findings provide important context to our results. Clement *et*
 66 *al.* conclude that at low CO_2 concentrations *T. pseudonana* utilizes a biophysical CCM through
 67 the active uptake CO_2 and bicarbonate, whereas at high CO_2 concentrations passive diffusion of
 68 CO_2 across membranes is sufficient to support photosynthesis. The authors provide evidence of
 69 strict regulation of CCMs depending on CO_2 concentrations. Additionally, they observed RuBisCO
 70 activity increased at high CO_2 , possibly resulting in higher CO_2 fixation capacity⁵. Unfortunately,
 71 our study does not have expression data for RuBisCO due to the poly(A) purification step during
 72 mRNA preparation, which bypasses chloroplast transcripts.

73 We cross-referenced expression changes of 36 transcripts across HC and LC conditions
 74 from our study with the corresponding CO_2 -responsive proteins reported in Table 1 of Clement *et*
 75 *al.* 2017. Transcript levels for most of these genes did not change significantly between HC and
 76 LC conditions in our study, with the exception of a small cluster of genes that were downregulated
 77 under HC conditions (**Fig. R4**). This CO_2 -responsive cluster consists of two carbon acquisition
 78 genes, a transmembrane protein (2078) and a delta carbonic anhydrase (34125). We have
 79 already identified these two genes as significantly downregulated in HC conditions (**see**
 80 **manuscript Supplementary Table S3**). The cluster also contained what Clement *et al.* labeled
 81 “Low CO_2 Inducible Protein of 63kDa” (LCIP63: 264181). The authors speculate that LCIP63 is
 82 not involved in general stress, is expressed only when CO_2 is limited, and might potentially play
 83 a wider role in how diatoms respond to CO_2 ⁶. In our study, LCIP63 is indeed significantly
 84 upregulated under LC conditions compared to HC conditions, but the transcript levels did not
 85 appear to respond to UVR stress. Additionally, it shows a diurnal pattern with higher expression
 86 during the light cycle. Our results are complementary with the findings of Clement *et al.*,
 87 supporting a potentially important role for LCIP63 in low CO_2 response (**Fig. R5**).

88

89 **CRITIQUE 3:** *The authors also should answer the following: Which resources would allow the*
 90 *cells to be more resilient (in addition to the CCM regulation).*

91 **RESPONSE:** It is important to be clear about what it means to be “more resilient”: “ecological
 92 resilience” is defined as the amount of disturbance that can be tolerated by a **system** without
 93 changing its state and still persist⁷⁻¹⁰. We can only speculate on which particular resources would
 94 allow the cells to be more resilient based on growth at high and low CO₂ concentrations. We
 95 characterized resilience at a phenotypic level by tracking growth characteristics (growth rate,
 96 carrying capacity, nitrate consumption, dynamics of response and recovery to UVR, and
 97 photosynthetic efficiency). We also took an unbiased approach towards investigating global
 98 physiological consequences by analyzing genome-wide transcriptome changes, in an attempt to
 99 discover plausible mechanisms that could have contributed to resilience and collapse.

100 However, based on previously published work we surmise a few potential processes that
 101 will be available to cells growing under HC conditions that may confer greater resilience. First, we

102 must consider the cost of activating CCMs. The
 103 process by which CCMs saturate RuBisCO to
 104 approximately 80%, is energy intensive as it
 105 requires transporting protons and inorganic
 106 carbon against a gradient across a membrane,
 107 while preventing diffusion of CO₂. This includes
 108 the energy investment into producing the catalytic
 109 machinery as well as its operating costs¹¹.
 110 Hopkinson et al. in 2011 defined the total
 111 energetic cost of operating a CCM¹² as, “the
 112 product of the energy expended to concentrate 1
 113 molecule of CO₂ at the site of fixation multiplied
 114 by the mole ratio of CO₂ transported to CO₂ fixed.”
 115 Their final calculations estimated that
 116 downregulation of CCMs in response to higher
 117 CO₂ concentrations would result in a 3-6% energy
 118 savings in diatoms. Wu et al. in 2014 performed
 119 an elegant experiment to test the consequence of
 120 such an energy savings on five diatom species.
 121 The authors observed that at increased CO₂
 122 concentrations photosystem II electron transport
 123 rates (ETR) were unaffected, nor was their elemental
 124 stoichiometry, but they did observe elevated
 125 growth rates¹³. Our observations are consistent
 126 with these calculations (see manuscript Fig. 1
 127 and 2) that growth rate and carrying capacities
 128 are higher for diatoms growing under HC
 129 conditions compared to LC conditions, particularly
 130 when cells were exposed to UVR (i.e., stage 2).
 131 Moreover, Wu et al. did not observe any
 132 difference in light capture dynamics and
 133 photosynthetically produced reductants under
 134 varying CO₂ concentrations, which led them
 135 to conclude that the enhanced growth is a
 136 result of increased diffusion rates of CO₂ and
 137 a lower metabolic cost due to downregulation
 138 of active carbon acquisition under elevated
 139 CO₂ conditions¹³.

132 Consistent with Wu et al’s
 133 findings, we did not observe
 134 significant difference in expression
 135 of 19 light harvesting complexes
 136 (LHC) between HC and LC
 137 conditions (Fig. R6). Another
 138 possible resource could come from
 139 reduced need for expending
 140 nitrogen in synthesizing CCMs
 141 under HC conditions; however,
 142 there was minimal differences
 143 in nitrate assimilation and urea
 144 cycle genes between HC and LC.
 145 Thus, we have not observed any
 146 evidence that the increased
 147 resilience under HC conditions
 148 could be attributed to any
 149 particular gene or pathway
 150 associated with stress response
 151 (oxidative stress response and
 152 DNA repair), light harvesting,
 153 and nitrogen metabolism.

Figure R5. Consistent over-expression of LCIP63 at low CO₂ during early or late phase of growth and in the light and dark cycles. Boxplots represent all transcripts throughout the stress-test. The insert panel shows its regulatory pattern throughout growth and with UVR exposure.

Figure R6. Bean plots of 19 light harvesting complexes. There was no significant difference in expression of LHCs at each timepoint ($n = 19$) between HC and LC conditions. Black bars indicate mean expression.

153 **CRITIQUE 4:** *The authors also should discuss known processes (not related to the CCM*
154 *(transporters) or FvFm) which are UV affected, potentially causing results as shown in the study.*

155 **RESPONSE:** High doses of UVR can damage proteins, DNA, and membranes directly or
156 indirectly through the production of ROS, inducing a system-wide response³. While we have
157 discussed some of these effects phenotypically vis-à-vis increased variability in photosynthetic
158 efficiency among replicates prior to collapse (**see manuscript Fig. 5**), we did not detect significant
159 differences (p-value < 0.05) in transcript levels of light harvesting complexes. As in Critique 1, Fig.
160 R3, we performed an additional differential expression analysis of 102 central carbon metabolism
161 genes that included genes of the pentose phosphate pathway, glycolysis, TCA cycle, and Calvin
162 cycle (**Fig. R7**) and did not observe any significant patterns. The downregulation of CCMs and
163 related genes under HC conditions remain the most probably process contributing to the results
164 of the study (**see manuscript Supplementary Table 3**). In summary, these additional analyses
165 did not reveal any conclusive evidence of a gene or process at the level differential regulation that
166 would indicate a significant potential influence, adding credence to our claim that collapse
167 occurred due to system failure, which manifested from loss of relational resilience. We have
168 updated the results and discussion on pages 8, 9, and 12, lines 179-192 and 278-282 to make
169 this point more explicit.

170
171 **CRITIQUE 5:** *Include more information on UV effects and how cells can cope with it. Maybe refer*
172 *to “Effects of ultraviolet radiation and CO2 increase on winter phytoplankton assemblages in a*
173 *temperate coastal lagoon (Domingues et al 2014).*

174 **RESPONSE:** The short-term mesocosm study performed by Domingues et al. 2014¹⁴ to evaluate
175 the effects of UVR and elevated CO₂ on phytoplankton community assemblages from a temperate
176 coastal lagoon are consistent with our findings. Interestingly, they observed an increase in diatom
177 abundance at high CO₂ conditions with UVR exposure. Diatom abundance changed from 14% of
178 the phytoplankton composition to 37%. The increase in diatom abundance over two days was
179 attributed mostly to a single *Thalassiosira* species. UVR did not have detrimental effects on the
180 dominant *Thalassiosira* centric diatom, but sensitivity to UVR is highly variable among
181 phytoplankton groups and species¹⁴. We have added insights from this study and other
182 information on UVR effects (please see manuscript pages 13 and 14, lines 301-303 and 306-307,
183 respectively).

184
185 **CRITIQUE 6:** *Additionally, please add information about effects of UV on RubisCO*
186 *(Vincent&Neale 2000 -which could affect the carbon metabolism).*

187 **RESPONSE:** Vincent and Neale's 2000 book chapter, "Mechanisms of UV damage to aquatic
188 organisms" has been a very helpful resource in regards to the direct and indirect effects of UVR
189 exposure³, which we have previously cited in our manuscript. We have summarized the net stress
190 imposed by UVR in the first paragraph of our response to critique 4. We were unable to determine
191 the expression level of Rubisco because the RNA library prep kit we used quantified transcripts
192 with poly(A) tails, and as a chloroplast gene, the Rubisco transcript does not have a poly(A) tail.
193 However, as mentioned in responses to critiques 3 and 4 we analyzed the differential gene
194 expression of central carbon metabolism (including Calvin cycle genes) and 19 light harvesting
195 complexes, with no clear or significant pattern of expression changes under HC versus LC
196 conditions. While we did not observe compelling changes in any specific gene or pathway, we did
197 uncover significant evidence that when cultures approached a point of collapse, there was lack of
198 coherence between the internal cellular state and the environment (i.e., loss of *relational*
199 *resilience*). Not only was this phenomenon predictive of collapse, but it also revealed that cultures
200 growing under HC conditions were able to maintain coherence between internal cellular state and
201 the environment over a longer timeframe, which we and many others^{12,14-17} speculate is because
202 of availability of more resources due to the conservation of energetic processes such as CCMs.

203

204 In Response to Reviewer #3:

205 **RESPONSE:** We take significant issue with some of the points made by Reviewer 3, which in our
206 opinion reflect a lack of understanding and appreciation of established dynamical systems theory
207 regarding behavior of complex systems on the verge of a critical transition. More importantly, we
208 are perplexed by the reviewer's discounting of the concept of *relational resilience*, which **IS** the
209 mechanism of population collapse. The reviewer wants a reductionist explanation for an emergent
210 systems-level phenomenon. It seems the reviewer wants a mechanistic explanation of how one
211 gene or process is contributing to collapse, when our data show in a compelling manner that the
212 underlying cause is systems failure resulting from discordance of the internal state of the cell and
213 its environmental context. Some of their suggestions and critiques are excessive and have gone
214 beyond the scope of this manuscript. Thus, we would like to request that the revised manuscript
215 not be sent back to Reviewer 3.

216 *Reviewer #3 Minor comment:*

217 *Figure 1 and Supplementary Figure 1 are nearly identical, with Supplementary Figure 1 being*
218 *more informative. Suggest reducing the redundancy of both and replace Figure 1 with*
219 *Supplementary Figure 1.*

220 **RESPONSE:** We updated **Fig. 1** of the manuscript to include nitrate consumption profiles.
221 However, we believe the updated **Supplementary Fig. 1** is also important as it shows growth
222 curves for each replicate culture.
223

224 REFERENCES:

- 225 1. Coesel, S. *et al.* Diatom PtCPF1 is a new cryptochrome/photolyase family member with
226 DNA repair and transcription regulation activity. *EMBO reports* **10**, 655–661 (2009).
- 227 2. Rijstenbil, J. W. Assessment of oxidative stress in the planktonic diatom *Thalassiosira*
228 *pseudonana* in response to UVA and UVB radiation. *Journal of Plankton Research* (2002).
- 229 3. Vincent, W. F. & Neale, P. J. Mechanisms of UV damage to aquatic organisms. In: de
230 Mora, S., Demers, S., Vernet, M. (Eds.), *The effects of UV radiation in the marine*
231 *environment*. Cambridge University Press, Cambridge, pp. 149–176 (2000).
- 232 4. Vernet, M. Effects of UV radiation on the physiology and ecology of marine
233 phytoplankton. In: de Mora, S., Demers, S., Vernet, M. (Eds.), *The effects of UV radiation*
234 *in the marine environment*. Cambridge University Press, Cambridge, pp. 237–278 (2000).
- 235 5. Clement, R., Dimnet, L., Maberly, S. C. & Gontero, B. The nature of the CO₂-concentrating
236 mechanisms in a marine diatom, *Thalassiosira pseudonana*. *New Phytologist* **209**, 1417–
237 1427 (2015).
- 238 6. Clement, R. *et al.* Responses of the marine diatom *Thalassiosira pseudonana* to changes
239 in CO₂ concentration: a proteomic approach. *Scientific Reports* 1–12 (2017).
- 240 7. Botton, S., van Heusden, M., Parsons, J. R., Smidt, H. & van Straalen, N. Resilience of
241 microbial systems towards disturbances. *Critical Reviews in Microbiology* **32**, 101–112
242 (2008).
- 243 8. Gunderson, L. H. Ecological resilience--in theory and application. *Annual review of ecology*
244 *and systematics* **31**, 425–439 (2000).
- 245 9. Holling, C. S. Resilience and stability of ecological systems. *Annual review of ecology and*
246 *systematics* (1973).
- 247 10. Griffiths, B. S. & Philippot, L. Insights into the resistance and resilience of the soil microbial
248 community. *FEMS Microbiol Rev* **37**, 112–129 (2013).
- 249 11. Raven, J. A., Beardall, J. & Giordano, M. Energy costs of carbon dioxide concentrating
250 mechanisms in aquatic organisms. *Photosynthesis Research* **121**, 111–124 (2014).
- 251 12. Hopkinson, B. M., Dupont, C. L., Allen, A. E. & Morel, F. M. M. Efficiency of the CO₂-

- 252 concentrating mechanism of diatoms. *Proc. Natl. Acad. Sci. U.S.A.* **108**, 3830–3837
253 (2011).
- 254 13. Wu, Y., Campbell, D. A., Irwin, A. J., Suggett, D. J. & Finkel, Z. V. Ocean acidification
255 enhances the growth rate of larger diatoms. *Limnol. Oceanogr.* **59**, 1027–1034 (2014).
- 256 14. Domingues, R. B., Guerra, C. C., Barbosa, A. B., Brotas, V. & Galvão, H. M. Effects of
257 ultraviolet radiation and CO₂ increase on winter phytoplankton assemblages in a
258 temperate coastal lagoon. *Journal of Plankton Research* **36**, 672–684 (2014).
- 259 15. Hennon, G. M. M. *et al.* Diatom acclimation to elevated CO₂ via cAMP signalling and
260 coordinated gene expression. *Nature Climate Change* **5**, 761–765 (2015).
- 261 16. Chrachri, A., Hopkinson, B. M., Flynn, K., Brownlee, C. & Wheeler, G. L. Dynamic changes
262 in carbonate chemistry in the microenvironment around single marine phytoplankton cells.
263 *Nature Communications* 1–12 (2017).
- 264 17. Giordano, M., Beardall, J. & Raven, J. A. *CO₂ concentrating mechanisms in algae:
265 mechanisms, environmental modulation, and evolution.* *Annu. Rev. Plant Biol.* **56**, 99–131
266 (2010).
- 267

REVIEWERS' COMMENTS:

Provided only confidential comments to the editor.